# The structural basis of the multi-step allosteric activation of Aurora B kinase

**Dario Segura-Peña**[1]*, **Oda Hovet**[1,2,3], **Hemanga Gogoi**[1],
**Jennine Dawicki-McKenna**[4], **Stine Malene Hansen Wøien**[1], **Manuel Carrer**[3,5],
**Ben E Black**[4], **Michele Cascella**[3,5]*, **Nikolina Sekulic**[1]*

[1]Centre for Molecular Medicine Norway (NCMM), Nordic EMBL Partnership, Faculty of Medicine, Oslo, Norway; [2]Department of Biosciences, University of Oslo, Oslo, Norway; [3]Hylleraas Centre for Quantum Molecular Sciences, University of Oslo, Oslo, Norway; [4]Department of Biochemistry and Biophysics, Penn Center for Genome Integrity, Epigenetics Institute, Perelman School of Medicine, University of Pennsylvania, Philadelphia, United States; [5]Department of Chemistry, University of Oslo, Oslo, Norway

**\*For correspondence:**
dario.segura-pena@ncmm.uio.no (DS-P);
michele.cascella@kjemi.uio.no (MC);
nikolina.sekulic@ncmm.uio.no (NS)

**Competing interest:** The authors declare that no competing interests exist.

**Abstract** Aurora B, together with IN-box, the C-terminal part of INCENP, forms an enzymatic complex that ensures faithful cell division. The [Aurora B/IN-box] complex is activated by autophosphorylation in the Aurora B activation loop and in IN-box, but it is not clear how these phosphorylations activate the enzyme. We used a combination of experimental and computational studies to investigate the effects of phosphorylation on the molecular dynamics and structure of [Aurora B/IN-box]. In addition, we generated partially phosphorylated intermediates to analyze the contribution of each phosphorylation independently. We found that the dynamics of Aurora and IN-box are interconnected, and IN-box plays both positive and negative regulatory roles depending on the phosphorylation status of the enzyme complex. Phosphorylation in the activation loop of Aurora B occurs intramolecularly and prepares the enzyme complex for activation, but two phosphorylated sites are synergistically responsible for full enzyme activity.

## Editor's evaluation

This important study investigates the dynamic activation mechanism of a key mitotic kinase complex, Aurora B/INCENP. The method of generating specifically phosphorylated forms of the complex is elegant, supporting a compelling experimental and computational analysis of how these sites synergistically activate Aurora B and providing insight into the dynamics underlying the activation mechanism. This work will be of interest to cell biologists and biochemists studying cell division and kinase regulation.

## Introduction

The chromosome passenger complex (CPC) is a complex of four proteins involved in the regulation of key mitotic events: correction of chromosome-microtubule attachment errors, assembly and stability of the spindle assembly checkpoint, and assembly and regulation of the contractile apparatus that drives cytokinesis (*Carmena et al., 2012*; *Vader et al., 2006*). The CPC is organized into two modules. The localization module consists of survivin, borealin, and the N-terminal part of INCENP, whereas the enzymatic module comprises Aurora B kinase and the C-terminal part of INCENP, called the IN-box. The enzymatic activity of Aurora B is essential for the regulatory functions of CPC (*Kelly and Funabiki, 2009*; *Krenn and Musacchio, 2015*; *Meraldi et al., 2004*), so Aurora B has been identified as an

attractive target for cancer chemotherapy (*Portella et al., 2011*), with several inhibitors patented in the last decades (*Jing and Chen, 2021*).

The enzymatic activity of the [Aurora B/IN-box] is tightly regulated by autophosphorylation. The activity of the complex increases by at least two orders of magnitude after the complex phosphorylates itself (*Zaytsev et al., 2016*). There are two well-defined autophosphorylation sites in the [Aurora B/IN-box] enzyme complex. One is a threonine in the Aurora B activation loop, and the second is located at the C-terminus of the IN-box and consists of two consecutive serine residues in the TSS motif (*Bishop and Schumacher, 2002*; *Honda et al., 2003*; *Sessa et al., 2005*).

Until recently, the molecular understanding of Aurora B autoactivation was based on a model derived from the structure of the partially phosphorylated enzyme, in which the threonine in the Aurora B activation loop was phosphorylated but the TSS motif of INCENP was either absent (*Sessa et al., 2005*; *Sessa and Villa, 2014*) or unstructured (*Elkins et al., 2012*). Recently, a structural study of the [Aurora C/IN-box] complex, which is specific for meiosis but otherwise very similar to the [Aurora B/IN-box] complex, provided valuable insight into the conformation the fully phosphorylated enzyme adopts (*Abdul Azeez et al., 2019*). The crystal structure shows that the activation loop of Aurora C and the IN-box TSS motif interact in the phosphorylated state and stabilize the extended (active) conformation of the activation loop. Although the enzyme complex was crystallized with an inhibitor bound to the ATP-binding site, it is likely that the observed interaction between the IN-box and the activation loop represents a conformation corresponding to the active state of the enzyme complex. Crystal structures are extremely valuable as high-resolution snapshots of enzyme complex, but they lack the dynamic information about the enzyme complex that is required for a complete understanding of the autoactivation process. In addition, there is not crystal structure available of the [Aurora B/IN-box] complex in the unphosphorylated state and therefore, there is an important gap in understanding the conformational changes taking place during the activation process.

Proteins are dynamic macromolecules that exist in multiple conformations in their native state. An organized protein will explore a smaller number of conformations with less structural variability, than a disordered protein that will explore a larger number of conformations in the same time. In addition, there is an increasing evidence that the dynamic properties of an enzyme play a critical role in its catalytic functions (*Ahuja et al., 2019*; *Tzeng and Kalodimos, 2012*). For example, an increase in dynamics or local flexibility may facilitate the adoption of the different conformations required during the catalytic cycle (*Boehr et al., 2006*; *Tzeng and Kalodimos, 2012*; *Xie et al., 2020*). On the other hand, the transition from a highly dynamic, disordered enzyme to a more structured enzyme has also been documented as a form of regulation of enzyme catalysis (*Wei et al., 2016*; *Xiao et al., 2015*). In particular, an intensive line of research on protein kinases has shown that the internal dynamics associated with the movements of the enzyme's subdomains regulate enzymatic activity (*Ahuja et al., 2019*; *Pegram et al., 2021*; *Taylor and Kornev, 2011*). While detailed dynamics analyses are available for several eukaryotic protein kinases activated by phosphorylation (*Foda et al., 2015*; *Wang et al., 2019*), including the closely related Aurora A (*Cyphers et al., 2017*; *Dodson and Bayliss, 2012*; *Gilburt et al., 2017*; *Pitsawong et al., 2018*; *Ruff et al., 2018*; *Tomlinson et al., 2022*), the effect of phosphorylation on the dynamic properties of the [Aurora B/IN-box] complex is not yet known.

Here, we use a combination of experimental (hydrogen-deuterium exchange coupled to mass-spectrometry [HDX-MS] and enzyme kinetics) and computational (molecular dynamics [MD]) approaches to follow the internal dynamics of the [Aurora B/IN-box] complex in solution - both in the unphosphorylated, inhibited, ([Aurora B/IN-box]$^{no-P}$) and fully phosphorylated, active, ([Aurora B/IN-box]$^{all-P}$) states. We found that in the absence of phosphorylation, there is structural disorder scattered along several regions of the enzyme complex. In contrast, in the fully phosphorylated enzyme complex, both Aurora B and the IN-box are structurally organized, resulting in a fully functional enzyme.

To investigate the contribution of individual phosphorylation sites to the dynamics and enzymatic activity, we used a protein ligation method to generate partially phosphorylated forms of the enzyme complex in which either only the activation loop of Aurora B ([Aurora B/IN-box]$^{loop-P}$) or only the TSS motif of the IN-box ([Aurora B/IN-box]$^{IN-P}$) is phosphorylated. Phosphorylation of each individual site in isolation has only a minor effect on enzymatic activity and structural stability, whereas simultaneous phosphorylation of the activation loop in Aurora B and the TSS motif in the IN-box synergistically affects the structure and dynamics of the enzyme complex, leading to a dramatic increase in kinase activity. Further, we identify phosphorylation in the activation loop of Aurora B as a rate-limiting

intramolecular event in autoactivation. Finally, only when both sites are phosphorylated are the global movements of the N- and C-lobes of the enzyme (opening and closing of the active site and twisting between the lobes) coordinated with the movements of the activation loop, resulting in a fully functional kinase.

Our combined experimental and theoretical approach has allowed us to study in parallel unphosphorylated, partially phosphorylated, and fully phosphorylated [Aurora B/IN-box] and to provide the missing information necessary to build a more detailed model of [Aurora B/IN-box] autoactivation. Our model is important for understanding the role of protein dynamics in achieving full enzyme activity, and it may also serve as a starting point for the development of new Aurora B-specific inhibitors.

## Results

### Autoactivation of [Aurora B/IN-box] by phosphorylation induces structural organization in multiple regions of the enzyme complex

The [Aurora B/IN-box] complex is activated by autophosphorylation, resulting in an enzyme with at least 100-fold higher enzymatic activity (*Zaytsev et al., 2016*; *Figure 1—figure supplement 1*). To obtain information on the possible structural and dynamic changes underlying the observed differences in enzymatic activity between unphosphorylated [Aurora B/IN-box]$^{no-P}$ and the fully phosphorylated enzyme complex, [Aurora B/IN-box]$^{all-P}$, we monitored the HDX in the protein backbone of [Aurora B/IN-box] from *Xenopus laevis*. HDX is based on the fact that in aqueous solution, the amide protons of the protein main chain are exchanged with protons from the solvent (*Mayne, 2016*). Thus, by changing the solvent from $H_2O$ to $D_2O$, HDX can be measured. The unstructured protein regions are the fast exchangers, while well-ordered regions where the amide backbone hydrogen atoms are involved in hydrogen bonds are exchanged more slowly. HDX analysis of proteins has proven useful in determining the activation mechanisms of numerous enzymes (*Dawicki-McKenna et al., 2015*; *Habibi et al., 2019*; *Lorenzen and Pawson, 2014*; *Sours and Ahn, 2010*; *Zhang et al., 2020*). By comparing the differences in deuterium uptake in the phosphorylated and unphosphorylated forms of the enzyme complex, we identified the regions of the enzyme where dynamic changes occur during phosphorylation. HDX measurements were performed using MS (Materials and methods).

We measured HDX in [Aurora B/IN-box]$^{no-P}$ and [Aurora B/IN-box]$^{all-P}$ at four time points (*Supplementary files 1 and 2*). Comparison of HDX measurements for the two phosphorylation states reveals several regions of the enzyme complex with profound HDX differences (*Figure 1* and *Figure 1—figure supplement 2*). The unphosphorylated form of the enzyme takes up more deuterium in all regions where HDX differences were detected, indicating greater flexibility and lower structural organization. Consequently, the fully active form, [Aurora B/IN-box]$^{all-P}$, has a lower HDX, indicating a more rigid structural organization. A detailed comparison between the two enzyme forms reveals several regions with significant HDX differences in Aurora B and in the IN-box (*Figure 1—figure supplement 2*; also indicated by black lines and corresponding secondary structure elements in *Figure 1A*). In Aurora B, there are four regions with high cumulative differences (labeled with numbered circles in *Figure 1A* and in *Figure 1—figure supplement 2B*). Two additional regions have statistically significant HDX differences, albeit relatively small. One region is at the N-terminus, spanning residues 93–120 and including β1-, β2-strands and the glycine loop, which is important for ATP binding. The other region is on the C-terminus, spanning αH and αI-helices.

In Aurora B, the first region with high ΔHDX spans residues 120–156 and includes part of the β3-strand, the αB-helix, the catalytic αC-helix, and the loop region following the αC-helix (*Figure 1A–C*). This portion of Aurora B includes the residues Aurora B$^{Lys122}$ and Aurora B$^{Glu141}$, which play a role in defining the active conformation of the enzyme (*Sessa et al., 2005*). The middle portion of the αC helix is particularly rigid in the phosphorylated enzyme complex (*Figure 1C*) and shows >15% HDX protection across all time points. It is known from studies of other eukaryotic protein kinases that the αC-helix plays a critical role in the organization of the productive enzyme complex (*Kornev and Taylor, 2015*). This helix contributes to the positioning of the activation segment, which includes the magnesium-binding loop (DFG loop), the activation loop with phosphorylated Aurora B$^{Thr248}$, and the P+1 loop (*Figure 1D*). The αC-helix residues also contribute to the formation of the R-spine, a series of hydrophobic interactions extending from the N-lobe to the C-lobe of protein kinases, that are critical for the adoption of an active conformation (*Kornev and Taylor, 2015*). In addition, helices αB

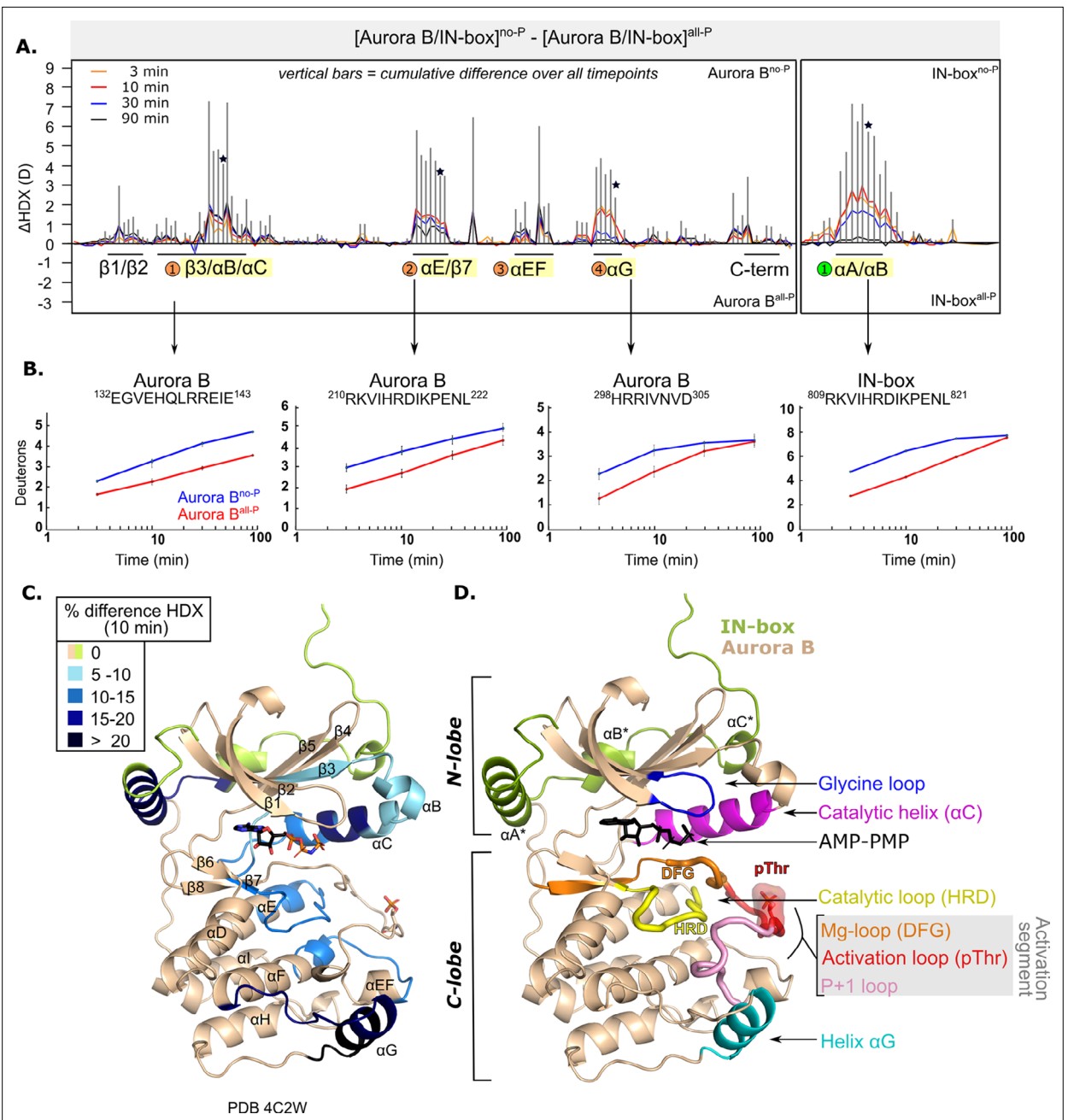

**Figure 1.** After phosphorylation, the [Aurora B/IN-box] complex becomes more rigid. (**A**) Butterfly-difference plot for a comparison of H/D exchange (HDX) differences between [Aurora B/IN-box]no-P and [Aurora B/IN-box]all-P, measured over four time points (orange - 3 min; red - 10 min; blue - 30 min; black - 90 min). Each of the vertical bars represents a single peptide arranged according to its position from the N- to the C-terminus. The size of the vertical bars represents the cumulative difference ([Aurora B/IN-box]all-P - [Aurora B/IN-box]no-P) or ΔHDX over all time points. It is noticeable that the difference in deuterium uptake is mainly positive, indicating an overall faster exchange and more flexible structure in [Aurora B/IN-box]no-P than in [Aurora B/IN-box]all-P for both Aurora B and IN-box. Secondary structural elements that are encompassed by peptides with significant differences in deuterium uptake are indicated below the baseline. Structural elements containing peptides with highest cumulative ΔHDX are highlighted in yellow and indicated with orange circles for Aurora B and green circle for the IN-box. The asterisk indicates representative peptides for which uptake plots are shown in (B) (**B**). The uptake plots for representative peptides with high uptake differences. (**C**) HDX differences between [Aurora B/IN-box]no-P and [Aurora B/IN-box]all-P captured after 10 min of exchange and mapped on ribbon diagram of the [Aurora B/IN-box] complex (PDB 4C2W). The Aurora B secondary structure elements are labeled. (**D**) Ribbon diagram of [Aurora B/IN-box] with Aurora B in beige and IN-box in green. The secondary structure elements of the IN-box (αA*, αB* and αC*) are labeled. The important catalytic and regulatory elements of Aurora B are highlighted: glycine loop (blue), catalytic helix αC (magenta), catalytic loop with histidine-arginine-aspartic acid (HRD) motif (yellow), and helix αG (cyan). The active segment consists of

*Figure 1 continued on next page*

*Figure 1 continued*

an Mg-loop with DFG motif (aspartic acid-phenylalanine-glycine) (orange), an activation loop with phosphorylated threonine (red), and a P+1 loop (pink) involved in peptide substrate binding.

The online version of this article includes the following figure supplement(s) for figure 1:

**Figure supplement 1.** Time course of product formation.

**Figure supplement 2.** Peptide coverage map and statistical analysis of deuterium uptake.

**Figure supplement 3.** EX1 kinetics observed in IN-box$^{\alpha A}$, peptide $^{807}$LTQAIRQQYYKPIDV$^{821}$, in [Aurora B/IN-box]$^{no-P}$, and [Aurora B/IN-box]$^{all-P}$.

**Figure supplement 4.** Mass photometry analysis for phosphorylated and unphosphorylated enzyme complex.

and αC provide an interacting surface for binding regulators of kinase activity in a variety of protein kinases (*Leroux and Biondi, 2020*) and are a site of interaction between the structurally similar Aurora A kinase and its binding partner TPX2 (*Bayliss et al., 2003*).

The second region spans residues 205–223 and includes the last five amino acids of the αE-helix, a linker region that includes the major catalytic loop (HRD loop) and the β7-strand. The third region spans residues 254–272 and includes the end of the activation segment, the αEF-helix, the loop upstream of the αF-helix, and the first turn of the αF-helix, which is an important organizer of the C-lobe (*Kornev et al., 2008*). The fourth region spans residues 284–305 and includes the αG-helix and the preceding loop. This region is part of the peptide substrate recognition site and is thought to play a regulatory role in protein kinases (*Ahuja et al., 2019*).

The peptides spanning the region containing the phosphorylated Aurora B$^{Thr248}$ are not shown in our coverage map (*Figure 1—figure supplement 2A*) because they did not pass the stringent peptide quality filter based on intensity and redundancy of the peptide. However, upon manual analysis, we did not detect any changes in deuterium uptake between the phosphorylated and unphosphorylated forms in this region. Deuterium exchange in this part of the protein (which is also observed in the peptides immediately upstream of Aurora B$^{Thr248}$, see *Supplementary file 5*) is very rapid, independently of enzyme phosphorylation, so that complete exchange occurs even at the earliest time points. This is in contrast to the second part of the activation loop, which includes the Aurora B$^{\alpha EF}$ helix (labeled region 3 in *Figure 1A*; peptide 254–260 in *Supplementary file 5*), where we clearly see HDX protection upon phosphorylation. It is possible that phosphorylation causes dynamic changes on a very fast scale (seconds or faster) in the part of the protein encompassing Aurora B$^{Thr248}$, but we could not detect them due to the limitation of our approach, which operates on the scale of minutes.

The IN-box is highly dynamic with a very fast HDX rate in both [Aurora B/IN-box]$^{no-P}$ and [Aurora B/IN-box]$^{all-P}$ (most peptides are fully exchanged at the early time points, *Supplementary file 5*). The only segment of IN-box where differences in deuteration are detected extends from positions 805 to 825 and contains the only well-defined secondary structure elements: IN-box$^{\alpha A}$ helix and the short IN-box$^{\alpha B}$ helix connected by a linker (*Figure 1A–D*). HDX kinetics for this region shows a strong component of EX1 kinetics and a smaller component of common EX2 kinetics. By definition, EX2 kinetics is common in the native folded state of proteins, while EX1 kinetics are less common in native conditions and is indicative of structural unfolding of local protein regions (*Arrington and Robertson, 2000*; *Chitta et al., 2009*; *Ferraro et al., 2004*; *Sperry et al., 2012*; *Walters et al., 2013*). In the IN-box$^{805-825}$ region, we observe the appearance of two clear isotopic envelopes (*Figure 1—figure supplement 3*), a hallmark of EX1 kinetics. The lower mass isotopic peak corresponds to the folded state of IN-box$^{805-825}$, where the amide-hydrogen bonds are more protected. The high mass isotope peak is indicative of the unfolded state of the IN-box$^{805-825}$ region, where multiple amide-hydrogen bonds break simultaneously and exchange hydrogen for deuterium. The EX1 behavior was observed in both [Aurora B/IN-box]$^{no-P}$ and [Aurora B/IN-box]$^{all-P}$, suggesting that IN-box$^{805-825}$ reversibly switches between the unfolded and folded states regardless of the phosphorylation state. However, the higher mass peak appears more rapidly in [Aurora B/IN-box]$^{no-P}$, suggesting that this region of IN-box visits the unfolded state more frequently in the unphosphorylated form of the enzyme complex. Overall, this observation is consistent with a more disordered IN-box in the unphosphorylated state.

A possible explanation for the large deuterium uptake in the unphosphorylated [Aurora B/IN-box] could be the dissociation of the IN-box from Aurora B at the low μM concentration (0.4 μM) used in our HDX experiment. In this scenario, the interacting surfaces of Aurora B and IN-box would be available for HDX. To determine whether the [Aurora B/IN-box] complex remained intact in our HDX

experimental conditions, we used mass photometry to measure the molecular weight (MW) of the enzymatic complex, [Aurora B/IN-box]$^{no-P}$ and [Aurora B/IN-box]$^{all-P}$ at low nM concentrations. In both samples, we observed a single uniform peak corresponding to MW of ~44 kD (MW$_{Aurora B}$=35.4 kD; MW$_{IN-box}$=8 kD), the measurements were performed at 10 nM enzyme concentration. This clearly confirms that the interactions between Aurora B and IN-box are stable under the HDX experimental conditions in the two states, phosphorylated and unphosphorylated, (*Figure 1—figure supplement 4*) and that the differences observed by HDX are due to phosphorylation-induced changes in enzyme dynamics rather than dissociation between the two polypeptide chains.

The higher degree of HDX protection detected in the active site and key regulatory regions of the enzyme (*Figure 1C and D*) suggests that the structural organization that occurs in the phosphorylated enzyme, [Aurora B/IN-box]$^{all-P}$, is important for enzymatic activity. Our results are consistent with similar studies on other protein kinases where phosphorylation of the threonine residue in the activation loop leads to rigidification of the structure (*Lorenzen and Pawson, 2014*; *Steichen et al., 2010*). To gain more detailed structural insight into the dynamics and structure of the [Aurora B/IN-box] complex and to better understand the specific role of the IN-box in regulating Aurora B kinase activity, we performed MD simulations.

## MD simulations reveal major conformational change in the IN-box upon enzyme complex phosphorylation

The HDX experiments show that phosphorylation has a clear effect on the dynamics of [Aurora B/IN-box], but do not provide information on the nature of the conformational and dynamic changes that take place. To gain insight into the structural rearrangements during the autoactivation process at atomic resolution, we performed a 2.8 µs MD simulation of the enzyme complex in the phosphorylated, [Aurora B/IN-box]$^{all-P}$, and the unphosphorylated, [Aurora B/IN-box]$^{no-P}$ states using the available X-ray structure (PDB 4C2W) (*Sessa and Villa, 2014*) as a template (see Materials and methods and *Figure 2*). This is a crystal structure of the *X. laevis* [Aurora B/IN-box] complex in which the activation loop is phosphorylated, and the IN-box contains residues 797–847 (ends just before the TSS motif). The structure contains an ATP analog (ANP-PMP) in the active site (instead of a bulky inhibitors) and represents an enzyme of the same species as our experimental system, so we consider it the most appropriate starting point for our MS simulations. The C-terminus of the IN-box was extended by the residues IN-box$^{848-856}$ (dashed line in the *Figure 2A*), to correspond exactly to our experimental enzyme complex, and the AMP-PMP in the active site was replaced by ATP -Mg. To obtain [Aurora B/IN-box]$^{no-P}$, the phosphate group of Aurora B$^{Thr248}$ was removed, whereas for [Aurora B/IN-box]$^{all-P}$, phosphate groups at IN-box$^{Ser849}$ and IN-box$^{Ser850}$ were added (*Figure 2A*; see Materials and methods for details). During the simulation, the systems were stable and retained the tertiary and quaternary structures in the folded regions of the complex (*Figure 2—figure supplement 1A*). The major structural differences between the two forms of the enzyme complex were observed in the conformation of the C-terminal part of IN-box and in the conformation of the activation loop.

The C-terminal region of the IN-box, which is not visible in the X-ray structure of Aurora B from *X. laevis* (*Sessa and Villa, 2014*) was first modeled as an unfolded coil in solution, pointing away from the enzyme body and not interacting with any part of the enzyme complex (*Figure 2A*, dashed line). In the MD simulation of [Aurora B/IN-box]$^{no-P}$, where neither Aurora B$^{Thr248}$ nor the TSS motif of IN-box are phosphorylated, the IN-box showed great flexibility. Its C-terminal part remains unstructured and does not stably interact with Aurora B throughout the simulation period (*Figure 2A* and *Video 1*).

On the contrary, the simulations of phosphorylated [Aurora B/IN-box] showed that the presence of phosphates in Aurora B and IN-box rapidly stabilizes a defined conformation in which the C-terminal region of IN-box interacts with the Aurora B activation loop (*Figure 2A* and *Video 2*). At the beginning of the simulation, two phosphates of IN-box (IN-box$^{pSer849}$ and IN-box$^{pSer850}$) are immediately stabilized by adjacent positively charged residues, IN-box$^{Arg843}$ and IN-box$^{Lys846}$. Moreover, IN-box$^{Arg847}$ forms an electrostatic interaction with the phospho-threonine of the activation loop, Aurora B$^{pThr248}$, early in the simulation (0.7 µs), and this interaction remains stable until the end of the simulation (the next 1.8 µs) (*Figure 2B and C*). In addition to IN-box$^{Arg847}$, Aurora B$^{pThr248}$ is stabilized by three arginine residues from Aurora B - two adjacent residues in the activation loop, Aurora B$^{Arg246}$ and Aurora B$^{Arg247}$, and Aurora B$^{Arg215}$ from the catalytic loop (HRD), closely resembling equivalent interactions in the Aurora C observed in [Aurora C/IN-box] crystal structure. Thus, stabilization of negative phosphate in

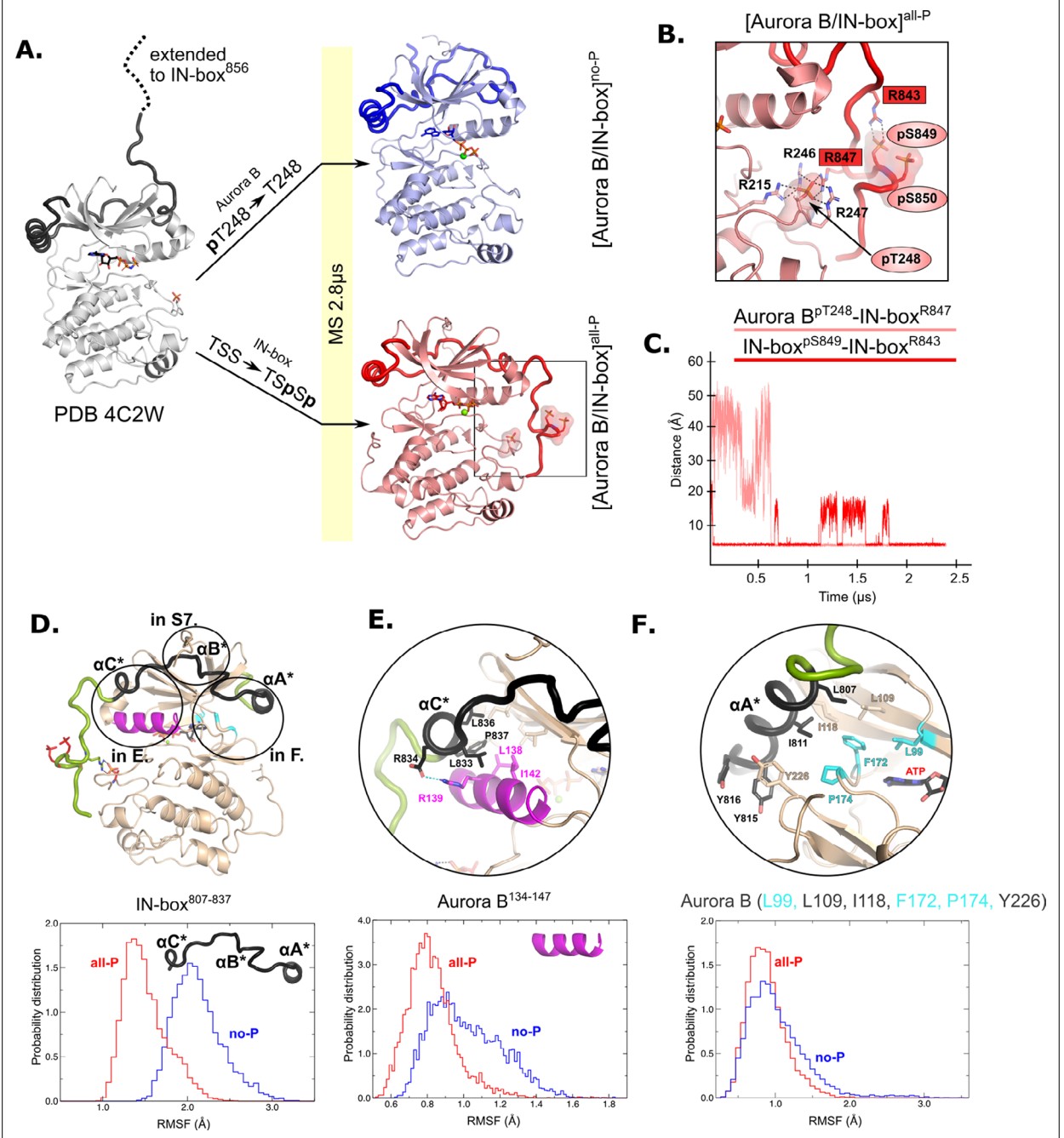

**Figure 2.** Molecular dynamics (MD) simulation of [Aurora B/IN-box]no-P and [Aurora B/IN-box]all-P reveals large conformational changes in the C-terminal part of IN-box and dynamic coupling between IN-box and the Aurora B catalytic helix. (**A**) Initial models for MD stimulation were generated from PDB 4C2W (ribbon diagram in gray; IN-box is shown as a black coil). The C-terminal part of IN-box was extended to the end (to include up to IN-box856; dotted line), AppNHp was replaced by ATP-Mg and for [Aurora B/IN-box]no-P, the phosphate group of Aurora BpT248 was removed. For [Aurora B/IN-box]all-P, phosphates were added to IN-boxSer849 and IN-boxSer850. Simulations for both phosphorylation states were run for 2.8 μs and the ribbon diagram for the final structure for [Aurora B/IN-box]no-P is shown in blue and the final structure for [Aurora B/IN-box]all-P is shown in red (IN-box is shown as coil in a darker color). The interactions within the boxed region of the [Aurora B/IN-box]all-P structure are shown in (B). (**B**) Enlarged view showing the interactions of Aurora BpThr248 with residues in Aurora B and IN-box in the final MD confirmation of [Aurora B/IN-box]all-P. (**C**) The distance between the phosphorus atom in Aurora BpThr248 and the C ζ atom in IN-boxArg847 (pink) and between the phosphorus atom in IN-boxpSer849 and the C ζ atom in IN-boxArg843 (red) during the MD simulation. The C ζ atom was chosen for the distance measurements because it is bonded to all three nitrogen atoms in the arginine residue, which can form a hydrogen bond with phosphate. (**D**) The ribbon diagram of the [Aurora B/IN-box]all-P structure (rotated 180° compared to the orientation in A). Aurora B is in beige and IN-box is in green. The N-terminal part of IN-box, IN-box807-837, which contains well-defined structural elements, αA*, αB*, αC*, is colored black. Parts of Aurora B that interact with this N-terminal IN-box and are important for catalysis are in magenta

*Figure 2 continued on next page*

*Figure 2 continued*

(Aurora B$^{\alpha C}$) and cyan (residues forming the ATP-binding site). The interfaces between three parts of the IN-box are circled and shown in more detail in panels (E, F) and *Figure 2—figure supplement 2*. Below is the plot of probability density in the y axis versus the root mean square fluctuations (RMSF) for IN-box$^{807-837}$. RMSF in [Aurora B/IN-box]$^{all-P}$ is in red and [Aurora B/IN-box]$^{no-P}$ is in blue. (E) Interface between IN-box$^{\alpha C*}$ and Aurora **B**. Below is the plot of probability density distribution versus RMSF for Aurora B$^{\alpha C}$ with same labeling as in (D). (F) Interface between IN-box$^{\alpha A*}$ and Aurora B. Three residues in Aurora B that form the ATP-binding site are cyan. Below is the same type of plot as is (D, E) for Aurora B (Leu99, Leu109, Ile118, Phe172, Pro174, and Tyr226).

The online version of this article includes the following figure supplement(s) for figure 2:

**Figure supplement 1.** Molecular dynamics analysis and comparison with crystal structures.

**Figure supplement 2.** The ribbon diagram of the [Aurora B/IN-box]$^{all-P}$ structure (oriented and colored as in *Figure 2*).

the [Aurora B/IN-box]$^{all-P}$ activation loop involves interaction with four arginine residues and is more extensive than phosphate stabilization in the initial structure (PDB 4C2W), in which Aurora B$^{pThr248}$ is present but the IN-box TSS motif is absent (*Figure 2—figure supplement 1B*). In the initial structure, Aurora B$^{pThr248}$ is stabilized by only two Aurora B arginines, Aurora B$^{Arg246}$ and Aurora B$^{Arg215}$, while Aurora B$^{Arg247}$ faces the solvent. An important common feature between our [Aurora B/IN-box]$^{all-P}$ simulation and the [Aurora C/IN-box]$^{all-P}$ crystal structure is that the IN-box interacts with the kinase activation loop stabilizing the active state of the kinase by reducing dynamics and favoring a more compact enzyme.

Overall, the structure from our [Aurora B/IN-box]$^{all-P}$ simulation and the [Aurora C/IN-box]$^{all-P}$ crystal structure are very similar. However, an important difference exists in the C-terminal IN-box region (*Figure 2—figure supplement 1C*). In the [Aurora B/IN box]$^{all-P}$ simulation, the C-terminal region of the IN-box is extended further toward the C-terminal lobe of the kinase and interacts with the phosphorylated activation segment, whereas in the [Aurora C/IN-box]$^{all-P}$ crystal structure there are reciprocal interaction in which arginines from the activation loop interact with the phosphorylated TSS motif of the IN-box. The latest interactions are not observed in our simulations. We suggest that the crystal structure of [Aurora C/IN-box]$^{all-P}$ is more likely to represent the final active conformational state of the enzyme complex and that the [Aurora B/IN-box]$^{all-P}$ from our simulation is more likely an intermediate state on the autoactivation trajectory. The transition from this intermediate state to the final state observed in the crystal structure could be a relatively slow process that occurs in time spans that we cannot capture with our simulation times. We discuss the possible reason for these differences in more detail in the Discussion.

## Experimental (HDX) and theoretical (MD) approaches imply that the dynamics of IN-box and Aurora B are interconnected

Both HDX and MD clearly show that the IN-box is highly disorganized in the unphosphorylated state of the enzyme complex. However, the large differences in the dynamics of the IN-box observed in

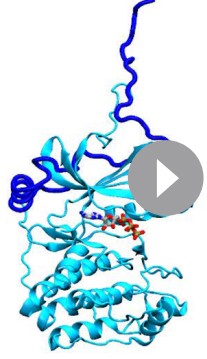

**Video 1.** 2.8 µs molecular dynamics (MD) simulation trajectory of [Aurora B/IN-box]$^{no-P}$.

https://elifesciences.org/articles/85328/figures#video1

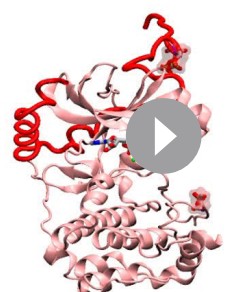

**Video 2.** 2.8 µs molecular dynamics (MD) simulation trajectory of [Aurora B/IN-box]$^{all-P}$.

https://elifesciences.org/articles/85328/figures#video2

MD simulations in the C-terminal part of the IN-box are not detectable in our HDX experiments. There could be two reasons for this apparent discrepancy. First, the C-terminal part of the IN-box is anchored to Aurora B only by side-chain interactions, whereas the main-chain hydrogens do not form stable hydrogen networks, leaving this region free for HDX despite a different conformation (*Figure 2—figure supplement 1D*). An alternative explanation is that HDX differences exist in the C-terminal region of the IN-box between two states, but the kinetics of the exchange occur in the millisecond range, which cannot be measured with our experimental setup.

Further, the N-terminal part, which has well-defined secondary structure elements, shows partial EX1 behavior, indicating partial unfolding (*Figure 1—figure supplement 3*). The N-terminal segment of the IN-box, which surrounds the N-lobe of Aurora B with multiple interactions, is the only part of the IN-box that is always associated with Aurora B. We wondered whether the dynamics of the interacting neighboring regions in Aurora B are coupled to the dynamics of the IN-box in the MD simulation.

We calculated the root mean square fluctuations (RMSFs) during MS simulations for the N-terminal part of the IN-box, IN-box$^{807-837}$, in the two forms of the enzyme complex (black coil in *Figure 2D*). Higher RMSF are associated with larger structural deformations. Consistent with the HDX results, the RMSFs for this region are significantly higher in [Aurora B/IN-box]$^{no-P}$ than in [Aurora B/IN-box]$^{all-P}$ (*Figure 2D*, graph). Next, we analyzed the RMSFs for the Aurora B regions interacting with the N-terminal fragment of the IN-box (*Figure 2E and F*, *Figure 2—figure supplement 2*). These regions include Aurora-B kinase elements essential for catalysis and substrate binding: the Aurora B$^{\alpha C}$, catalytic helix (*Figure 2E*, magenta), and the region forming the ATP-binding site (*Figure 2F*, cyan), respectively. In the dephosphorylated state, Aurora B$^{\alpha C}$ exhibits a very broad RMSF distribution shifted toward higher values, consistent with strong helix deformations. In contrast, the RMSF values in [Aurora B/IN-box]$^{all-P}$ show a narrow distribution, clearly indicating a more structured Aurora B$^{\alpha C}$. Although the difference is more subtle in the region of Aurora B that forms the ATP-binding site and is in contact with IN-box$^{\alpha A}$ (*Figure 2F*), it follows the same trend indicating structural stiffening upon phosphorylation and more disorder in the non-phosphorylated state. In summary, the Aurora B regions in contact with the N-terminal IN-box segment are structurally more disordered in the absence of phosphorylation than when the enzyme complex is fully phosphorylated, strongly suggesting that the dynamics of the adjacent regions of Aurora B and IN-box are coupled. In the unphosphorylated form of the enzyme complex the structural disorder spreads from one partner to the other. Therefore, the IN-box could act as a negative regulator of the kinase activity by transferring disorder to the kinase domain. Conversely, structural order is imposed on both Aurora B and the IN-box by phosphorylation.

However, the [Aurora B/IN-box]$^{all-P}$ complex is phosphorylated in both Aurora B (activation loop - Aurora B$^{Thr248}$) and the IN-box (TSS motif - IN-box$^{pSer849}$ and IN-box$^{pSer850}$), so it is not clear whether the observed changes are the result of phosphorylation of the Aurora B activation loop or the TSS motif in the IN-box, or both. We next wanted to determine the contribution of each of these phosphorylation sites to the dynamics of the enzyme complex and to its enzymatic activity.

## Generation of partially phosphorylated [Aurora B/IN-box] complexes

Previous attempts to uncouple the effects of the two phosphorylation sites in the [Aurora B/IN-box] enzyme complex were based on the analysis of phospho-null mutants (*Abdul Azeez et al., 2019*; *Wang et al., 2011a*) in which the TSS motif was changed to TAA. Although the mutational approach to mimic a phospho-null IN-box was possible, phospho-mimetic mutations in which the threonine in the activation loop or the serine residues in the IN-box TSS motif are replaced by glutamic acids to mimic phosphorylation resulted in catalytically dead enzymes, contrary to expectations. Phospho-null mutations in the activation loop also resulted in a dead kinase. Since it was not possible to obtain enzyme complexes with phosphorylation only in the IN-box, the contribution of phosphorylation of the activation loop to enzyme activity could not be understood.

Because the IN-box is relatively close to the C-terminal end of the construct, this region of the protein is suitable for protein ligation. Therefore, we used a synthetic peptide approach to introduce post-translational modifications (*Ghosh et al., 2011*; *Figure 3A*). We generated a truncated IN-box$^{790-844}$ (IN-box$^{\Delta C}$) fused to an intein that was co-expressed with Aurora B in a similar manner to the original [Aurora B/IN-box] construct. After purification of the complex and intein cleavage, the resulting [Aurora B/IN-box]$^{IN-\Delta C}$ was ligated with a synthetic peptide containing the IN-box residues IN-box$^{845-858}$ including the TSS motif (*Figure 3B*). The resulting protein differs from the original construct

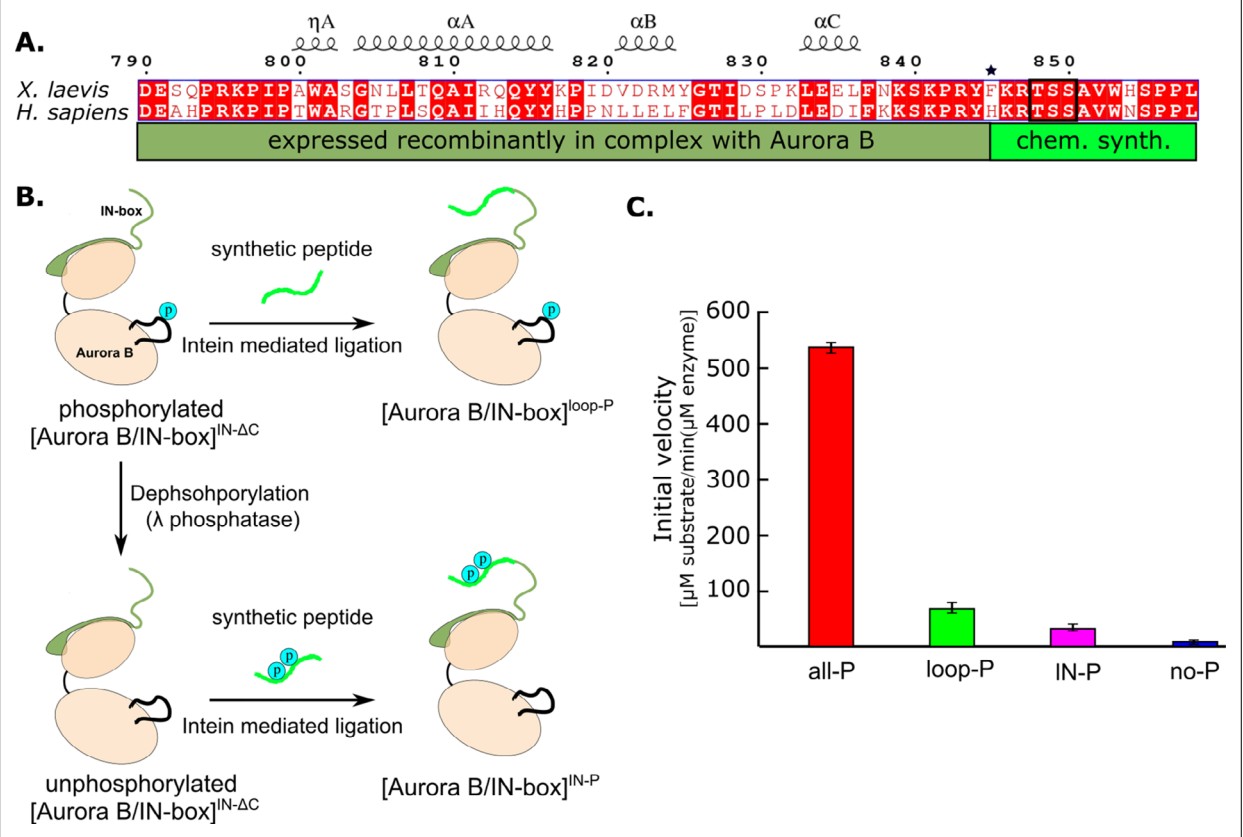

**Figure 3.** Design and kinetic analysis of the partially phosphorylated [Aurora B/IN-box] complex. (**A**) Sequence overlay of the IN-box, region of INCENP, from *Homo sapiens* and *X. laevis*. The conserved amino acids are shown in red and the elements of the secondary structure are at the top. Residues 790–844 (dark green) are expressed together with Aurora B, in a bicistronic construct, giving rise to [Aurora B/IN-box]$^{IN-\Delta C}$, which is then ligated to a synthetic peptide containing residues 846–858 (light green). The synthetic peptide contains a TSS motif that can be phosphorylated (black box). The Aurora B$^{Phe845}$ (indicated by asterisks) is converted to Cys during ligation. (**B**) The schematic shows the steps to prepare [Aurora B/IN-box]$^{loop-P}$ and [Aurora B/IN-box]$^{IN-P}$ using [Aurora B/IN-box]$^{IN-\Delta C}$ and synthetic peptides. (**C**) Enzymatic activity determined from initial velocities for [Aurora B/IN-box]$^{all-P}$ (red), [Aurora B/IN-box]$^{loop-P}$ (green), [Aurora B/IN-box]$^{IN-P}$(magenta), and [Aurora B/IN-box]$^{no-P}$ (blue).

The online version of this article includes the following source data and figure supplement(s) for figure 3:

**Figure supplement 1.** SDS-PAGE gel showing purification of partially phosphorylated [Aurora B/IN-box] enzymes.

**Figure supplement 1—source data 1.** Original uncropped gels shown in *Figure 3—figure supplement 1*.

**Figure supplement 2.** Mass spectra confirming presence or absence of phosphates in peptides Aurora B (248-264) and IN-box (848-858) for different phosphorylaton forms of the complex.

**Figure supplement 3.** Kinetic analysis of partially phosphorylated enzyme complex forms.

by a single amino acid that remains as a 'scar' of the ligation - IN-box$^{Phe845}$ becomes IN-box$^{Cys845}$ (indicated by the asterisk in *Figure 3A*). The enzyme complex containing phosphorylated Aurora B$^{Thr248}$ in the activation loop and the unphosphorylated IN-box, [Aurora B/IN-box]$^{loop-P}$, was obtained by ligation of the [Aurora B/IN-box]$^{IN-\Delta C}$ with a peptide containing the unphosphorylated IN-box$^{845-858}$ (*Figure 3B*). To ensure that Aurora B$^{Thr248}$ was phosphorylated, the [Aurora B/IN-box]$^{IN-\Delta C}$-intein was incubated with traces of the wild-type [Aurora B/IN-box]$^{all-P}$ complex (wt) before ligation. The wt enzyme was removed in a purification step before ligation (see Materials and methods for further details). To generate the enzyme complex phosphorylated only in the TSS motif of the IN-box, [Aurora B/IN-box]$^{IN-P}$, the truncated [Aurora B/IN-box]$^{IN-\Delta C}$-intein complex, was incubated with $\lambda$-phosphatase to ensure that Aurora B$^{Thr248}$ was not phosphorylated. After further purification to remove the $\lambda$-phosphatase, the unphosphorylated [Aurora B/IN-box]$^{IN-\Delta C}$ was ligated with an IN-box$^{845-858}$ phosphorylated at the two serine residues of the TSS motif. In all cases, size exclusion chromatography was performed as a final purification step to ensure that unligated peptides were removed (*Figure 3—figure supplement 1*),

whereas the presence or absence of phosphates in the Aurora B$^{Thr248}$ or TSS motif in the IN-box was confirmed by MS (*Figure 3—figure supplement 2*).

Having prepared semisynthetic, partially phosphorylated intermediates, we wanted to test whether they could reach their full enzymatic activity when autophosphorylated in the presence of ATP-Mg. Indeed, both semisynthetic constructs showed similar initial rates of enzymatic activity as the wt [Aurora B/IN-box]$^{all-P}$ after incubation with ATP-Mg (*Figure 3—figure supplement 3A*). This gave us confidence that neither the single mutation nor the process of chemical ligation affected the intrinsic nature of the enzyme complex.

## The two phosphorylation sites synergistically contribute to Aurora B kinase activity

Next, we compared the initial rates of enzymatic reaction for the partially phosphorylated enzyme complexes with fully phosphorylated, [Aurora B/IN-box]$^{all-P}$, and unphosphorylated, [Aurora B/IN-box]$^{no-P}$, in the presence of high substrate peptide concentrations (600 µM) (*Figure 3C*). Interestingly, although both partially phosphorylated enzyme complexes showed detectable enzymatic activity, their initial rates were much lower than those of the fully active enzyme, [Aurora B/IN-box]$^{all-P}$. [Aurora B/IN-box]$^{loop-P}$ exhibits only ~10% and [Aurora B/IN-box]$^{IN-P}$ only ~5% of the maximum initial rate. Although these values are still higher than the 1% activity measured for the unphosphorylated enzyme, [Aurora B/IN-box]$^{no-P}$, they are drastically lower than the maximum initial rates measured for the fully phosphorylated enzyme complex, [Aurora B/IN-box]$^{all-P}$. Thus, the simple addition of the activities of [Aurora B/IN-box]$^{loop-P}$ and [Aurora B/IN-box]$^{IN-P}$ accounts for only ~15% of the full enzyme activity, clearly showing that the phosphorylation of the Aurora B activation loop and the TSS motif of the IN-box act synergistically, rather than independently, to achieve the full activity of the [Aurora B/IN-box] complex.

For the [Aurora B/IN-box]$^{loop-P}$ construct, we determined the kinetic constants, $v_{max}$ and $K_M$, for the peptide substrate (*Figure 3—figure supplement 3B*). The $v_{max}$ of this enzyme complex is about three-fold lower than that of the fully phosphorylated, [Aurora B/IN-box]$^{all-P}$, complex, whereas $K_M$ increases dramatically (10-fold) (*Table 1*). This result suggests that the absence of phosphates in the IN-box TSS motif has a strong effect on substrate binding and a moderate effect on catalysis of the reaction and supports the role of the IN-box in stabilizing the open conformation of the activation segment that forms the peptide-substrate binding site. Overall, absence of phosphates in IN-box results in a 20-fold decrease in the catalytic efficiency ($k_{cat}/K_M$) of [Aurora B/IN-box]$^{loop-P}$ compared to the fully phosphorylated complex. Similar kinetic behavior was observed by *Abdul Azeez et al., 2019*, in a construct where TSS was replaced with TAA.

We attempted to determine kinetic constants for [Aurora B/IN-box]$^{IN-P}$, but this was not possible because a saturating concentration of the substrate peptide could not be obtained for this partially phosphorylated enzyme complex.

In summary, the experiments with the phosphorylated intermediates allowed us to unambiguously determine the contribution of each phosphorylation site independently and conclude that the

**Table 1.** Kinetic constants for [Aurora B/IN-box] complexes.

| Construct | $k_{cat}$ (s$^{-1}$) | $K_M$ (µM) | $k_{cat}/K_M$ (M$^{-1}$ s$^{-1}$) |
|---|---|---|---|
| [Aurora B/IN-box]$^{all-P}$ | 12 | 179 | $7 \times 10^4$ |
| [Aurora B/IN-box]$^{loop-P}$ | 4 | 1208 | $0.3 \times 10^4$ |
| [Aurora B/IN-box]$^{IN-P}$ | ~0.5* | ND | ND |
| [Aurora B/IN-box]$^{no-P}$ | ~0.08* | ND | ND |
| [Aurora B/IN-box$^{Arg843Ala}$] | 7 | 260 | $3 \times 10^4$ |
| [Aurora B/IN-box$^{Arg847Ala}$] | 16 | 860 | $1.9 \times 10^4$ |

ND - The $K_M$ value could not be determined due to the very high concentrations of peptide substrate required to produce measurable changes in the enzyme velocities.

*The reported $k_{cat}$ is estimated based on the $v_{max}$ values obtained by a non-linear fit to the Michaelis-Menten equation, using the GraphPad Prism software.

phosphorylations of the Aurora B activation loop and the IN-box sites contribute synergistically to the kinase activity.

## Phosphorylation of each subunit of the complex leads to partial rigidification of the entire complex, but only the fully phosphorylated enzyme complex achieves complete structuring

We wanted to investigate whether the impaired catalytic activity in the partially phosphorylated enzyme complexes correlates with their structural organization, and therefore measured HDX in the [Aurora B/IN-box]$^{loop-P}$ and the [Aurora B/IN-box]$^{IN-P}$ enzyme complexes (*Supplementary files 3 and 4*).

Because our partially phosphorylated complexes have the IN-box$^{Phe845Cys}$ mutation, we first asked whether [Aurora B/IN-box$^{Phe845Cys}$] exhibits the same HDX trends as [Aurora B/IN-box] when unphosphorylated and fully phosphorylated. From now on, [Aurora B/IN-box$^{Phe845Cys}$] in fully phosphorylated and unphosphorylated forms will be referred to as [Aurora B/IN-box]$^{all-P*}$ and [Aurora B/IN-box]$^{no-P*}$, respectively. Indeed, [Aurora B/IN-box]$^{all-P*}$ and [Aurora B/IN-box]$^{no-P*}$ show similar HDX pattern as the enzyme complex without the mutation (*Figure 4—figure supplement 1*). We also observe EX1 behavior in the N-terminal part of the IN-box (*Figure 4—figure supplement 2*).

Both partially phosphorylated enzyme forms exhibited intermediate levels of HDX protection (*Figure 4*, *Figure 4—figure supplement 3* and *Figure 4—figure supplement 4*). Each of the phosphorylations alone results in intermediate structuring of the entire enzyme complex compared with the fully phosphorylated enzyme. For example, two Aurora B regions defining the catalytic site, the catalytic helix (Aurora B$^{αC}$) and the catalytic loop (HRD loop), each exhibit partial H/D protection in both partially phosphorylated forms of the enzyme complex, corresponding to approximately half of the full protection observed in the fully phosphorylated active enzyme complex (*Figure 4B*, first two regions). The only exceptions to this rule are two regions that show equal or even greater H/D protection upon partial phosphorylation (black boxes in *Figure 4A* and *Figure 4B*, last two peptides). In the [Aurora B/IN-box]$^{IN-P}$ complex, only the IN-box$^{αA}$ helix shows high HDX protection. Interestingly, rigidification of the IN-box$^{αA}$ only partially affects neighboring Aurora B parts (see *Figure 4A and B*, first region, encompassing Aurora B$^{αC}$), suggesting that structuring of the IN-box is not sufficient to cause complete rigidification of the neighboring Aurora B regions. In the [Aurora B/IN-box]$^{loop-P}$ complex, only the Aurora B$^{αG}$-helix is strongly protected in HDX experiments (*Figure 4B*). We further confirmed that this rigidification is independent of phosphorylation of the TSS motif in the IN-box, by measuring HDX in the enzyme complex in which TSS is mutated to TAA (*Figure 4—figure supplement 5*).

Thus, HDX analysis shows that phosphorylation in the activation loop and phosphorylation in the IN-box each individually affect the entire enzyme complex, but full structural stability is achieved only when both units are phosphorylated, consistent with the synergistic interaction inferred from enzyme kinetics experiments.

## Phosphorylation of the Aurora B activation loop triggers a conformational change in the C-terminal region of the IN-box

To understand how partial phosphorylation affects enzyme structure, we next performed MD simulations for partially phosphorylated enzyme complexes in the same manner as for unphosphorylated and fully phosphorylated [Aurora B/IN-box] (*Figure 5A* and *Figure 5—figure supplement 1A*).

In [Aurora B/IN-box]$^{IN-P}$, the phosphorylated TSS motif was rapidly stabilized by adjacent positively charged residues (IN-box$^{Lys839}$, IN-box$^{Lys841}$, and IN-box$^{Arg847}$), albeit in a different conformation than in [Aurora B/IN-box]$^{all-P}$ (*Figure 5—figure supplement 1B* and *Video 3*). The phosphorylated IN-box interacts early in the simulation with the positively charged patch in the N-lobe of Aurora B and never extends to the C-lobe of Aurora B, in contrast to [Aurora B/IN-box]$^{all-P}$. Therefore, in [Aurora B/IN-box]$^{IN-P}$, the phosphorylated IN-box remains stably associated only with the N-lobe for the duration of the MD simulation. The stabilization of the N-terminal part of IN-box in this intermediate, that we observe with HDX, may be due to the interaction of the C-terminal IN-box with the N-terminal lobe of Aurora B. This interaction results in a more rigid IN-box, albeit in a conformation that is unproductive for enzyme activity.

A very interesting conformational change was observed in [Aurora B/IN-box]$^{loop-P}$, where only the activation loop of Aurora B is phosphorylated, the C-terminal region of IN-box is immediately

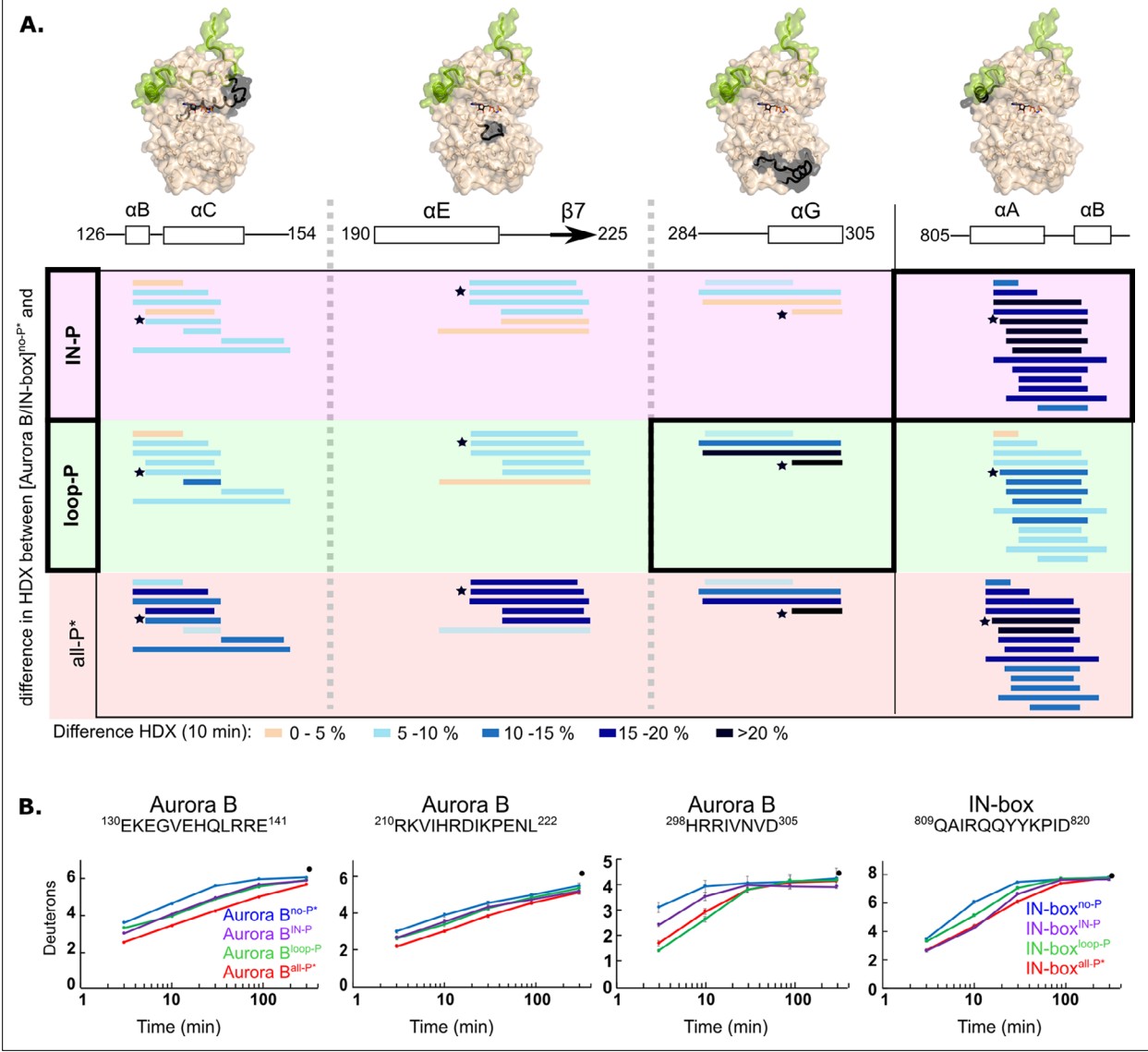

**Figure 4.** Two phosphorylations are required for maximal hydrogen-deuterium exchange (HDX) protection in [Aurora B/IN-box] with only two exceptions in Aurora B$^{\alpha G}$ and IN-box$^{\alpha A-\alpha B}$. (**A**) Surface representation of the [Aurora B/IN-box] molecule (PDB 4C2W), with regions where HDX differences were detected shown in black. A difference plot of HDX uptake between [Aurora B/IN-box]$^{no-P*}$ and either [Aurora B/IN-box]$^{loop-P}$ (second row) or [Aurora B/IN-box]$^{IN-P}$ (first row) after 10 min is shown for peptides in three different regions of Aurora B and one region in IN-box (these regions show the largest differences and are the same as in *Figure 1*). For reference, the differences in HDX uptake between [Aurora B/IN-box]$^{no-P*}$ and [Aurora B/IN-box]$^{all-P*}$ are shown in the third row (they both carry the same 'scar' mutation as partially phosphorylated intermediates to calibrate for the effects of the mutation on enzyme dynamics). Peptides are shown as horizontal lines and colored according to differences in deuterium uptake (see legend below). The asterisks indicate representative peptides for which uptake plots are shown in (**B**). Note that in [Aurora B/IN-box]$^{loop-P}$ only Aurora B$^{\alpha G}$ shows HDX protection comparable to [Aurora B/IN-box]$^{all-P}$, while in [Aurora B/IN-box]$^{IN-P}$ this is only true for IN-box$^{\alpha A}$. (**B**) The uptake plots for representative peptides from (**A**). The black dot is showing HDX for fully deuterated peptide under same experimental condition.

The online version of this article includes the following figure supplement(s) for figure 4:

**Figure supplement 1.** HDX comparison of unphosphorylated and fully phosphorylated [Aurora B/IN-box] complex where IN-box carries the **Phe845Cys** mutation.

**Figure supplement 2.** EX1 kinetics observed in IN-box$^{\alpha A}$, peptide $^{807}$LTQAIRQQYYKPIDV$^{821}$, in [Aurora B/IN-box$^{Phe845Cys}$]$^{no-P}$ and [Aurora B/IN-box$^{Phe845Cys}$]$^{all-P}$.

**Figure supplement 3.** HDX comparison of unphosphorylated [Aurora B/IN-box] with the IN-box mutation Phe845Cys and the [Aurora B/IN-box] **IN-P** intermediate.

**Figure supplement 4.** Differences in deuterium uptake between [Aurora B/IN-box]$^{no-P*}$ and [Aurora B/INbox]$^{loop-P}$.

**Figure supplement 5.** Structuring of Aurora B$^{\alpha G}$ is independent of phosphorylation in the IN-box TSS motif.

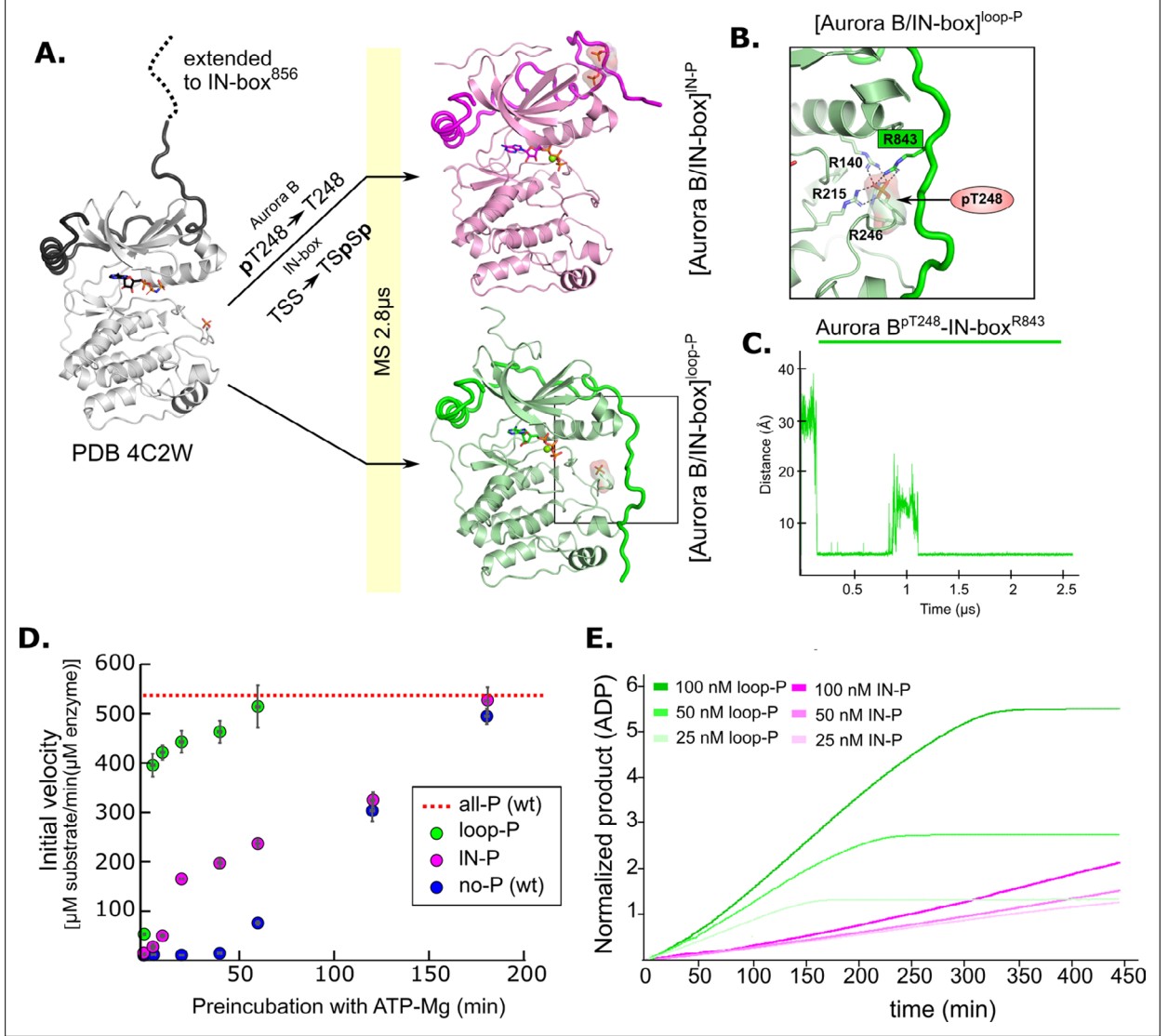

**Figure 5.** Molecular dynamics (MD) simulation and kinetic experiments show that phosphorylation of the Aurora B activation loop triggers a conformational change in the C-terminal region of the IN-box priming the [Aurora B/IN-box] complex for the activation. (**A**) Initial models for MD stimulation were generated as previously described (see legend to *Figure 2*). For [Aurora B/IN-box]$^{IN-P}$, the phosphate group of Aurora B$^{pT248}$ was removed, and phosphates were added to IN-box$^{Ser849}$ and IN-box$^{Ser850}$. For [Aurora B/IN-box]$^{loop-P}$, no phosphates needed to be added or removed because the initial structure already contained Aurora B$^{pThr248}$. Simulations were run for 2.8 μs and ribbon diagram of the final structure for [Aurora B/IN-box]$^{IN-P}$ is shown in magenta and the final structure for [Aurora B/IN-box]$^{loop-P}$ is shown in green (IN-box is shown as a coil in a darker color). The interactions, within the boxed region of the [Aurora B/IN-box]$^{loop-P}$ structure, are shown in (B). (**B**) Enlarged view showing the interactions of Aurora B$^{pThr248}$ with residues in Aurora B and IN-box in the final MD confirmation of [Aurora B/IN-box]$^{loop-P}$. (**C**) The change in the distance between the phosphorus atom in Aurora B$^{pThr248}$ and the C ζ atom in IN-box$^{Arg843}$ during the MD simulation. (**D**) Autoactivation assay for unphosphorylated or partially phosphorylated complexes. The enzyme was first preincubated with ATP-Mg at different times, followed by an enzymatic assay to determine the initial rate of substrate peptide phosphorylation. The plot shows the initial rates of enzymatic activity as a function of incubation time with ATP for [Aurora B/IN-box]$^{no-P}$ (blue), [Aurora B/IN-box]$^{loop-P}$ (green), and [Aurora B/IN-box]$^{IN-P}$ (magenta). The full enzymatic activity of [Aurora B/IN-box]$^{all-P}$ is shown as a red dotted line. Initial velocity measurements after different ATP preincubation times were performed in three independent experiments, and the standard deviation is indicated for each time point. (**E**) Dodson kinetic test. Normalized product release curves for [Aurora B/IN-box]$^{loop-P}$ (green) and [Aurora B/IN-box]$^{IN-P}$ (magenta) at 25, 50, and 100 nM concentration in the presence of the peptide substrate and ATP. Note that the pink traces overlap during the first ~50 min of incubation, indicating that phosphorylation of the activation loop is an intramolecular, concentration-independent, reaction.

The online version of this article includes the following figure supplement(s) for figure 5:

**Figure supplement 1.** Analysis of MD for [Aurora B/IN-box] phosphorylated intermediates.

**Figure supplement 2.** Phosphorylation in the activation loop is the intramolecular initial step in activation.

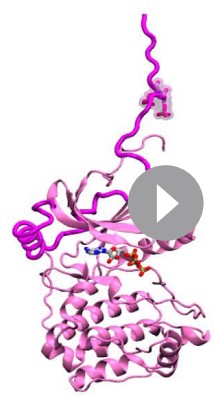

**Video 3.** 2.8 µs molecular dynamics (MD) simulation trajectory of [Aurora B/IN-box]^IN-P^.
https://elifesciences.org/articles/85328/figures#video3

attracted to the Aurora B^pThr248^ (*Figure 5* and *Video 4*), in a similar manner as in [Aurora B/IN-box]^all-P^. In our simulation, we observe that Aurora B^pThr248^ retains the same hydrogen bonds as in the starting crystal structure of Aurora B (PDB 4C2W), in addition to Aurora B^Arg140^ also being recruited from the Aurora B^αC^-helix (*Figure 5B*). Interestingly, an electrostatic interaction between Aurora B^pThr248^ and the IN-box^Arg843^ occurred after only 0.25 µs (which is even faster than a similar interaction in the case of [Aurora B/IN-box]^all-P^) and persisted throughout the rest of the simulation time (*Figure 5C*). Finally, due to this large conformational change, the IN-box forms several other electrostatic and hydrophobic interactions along the N- and C-lobes of the enzyme. The final structure of [Aurora B/IN-box]^loop-P^ at the end of the simulation resembles [Aurora B/IN-box]^all-P^ (*Figure 2* and *Figure 5*). This was a puzzling result, because both kinetics and HDX indicated significantly lower activity and a higher degree of disorder in [Aurora B/IN-box]^loop-P^ than in [Aurora B/IN-box]^all-P^. This unexpected result prompted us to perform a more detailed kinetic analysis.

## Phosphorylation of the activation loop is the intramolecular rate-limiting step in the activation process

Both partially phosphorylated complexes have low enzymatic activity when only one of the two sites is phosphorylated. They can recover their full enzymatic activity when the other site is phosphorylated during the autophosphorylation process (*Figure 3—figure supplement 3A*). We wanted to know whether one of the partially phosphorylated complexes can autoactivate faster than the other.

We preincubated [Aurora B/IN-box]^no-P^ (wt), [Aurora B/IN-box]^IN-P^, and [Aurora B/IN-box]^loop-P^ with ATP-Mg and monitored their autoactivation by determining the initial velocities after different preincubation times. We found that all three complexes can achieve the same activity as [Aurora B/IN-box]^all-P^ (wt) (*Figure 3—figure supplement 3A*), but the autoactivation proceeds with different kinetics (*Figure 5D*). As expected, the [Aurora B/IN-box]^no-P^, devoid of any phosphorylation, autoactivates slowest. However, the [Aurora B/IN-box]^loop-P^ is autoactivated much faster than the [Aurora B/IN-box]^IN-P^.

Previously, Aurora B autoactivation by phosphorylation was found to occur in two steps: first a slow intramolecular step followed by a faster intermolecular step (*Zaytsev et al., 2016*) but the tools to unambiguously correlate the kinetic steps with the specific phosphorylation events during autoactivation were not available. Our autoactivation experiments with partially phosphorylated intermediates clearly show that phosphorylation of the Aurora B activation loop is the slow step, as [Aurora B/IN-box]^IN-P^ (where Aurora B^Thr248^ is not phosphorylated) is slowly autoactivated. On the other hand, phosphorylation of IN-box is the fast step because [Aurora B/IN-box]^loop-P^ (where IN-box is not phosphorylated) autoactivates quickly. To further confirm that phosphorylation of the Aurora B activation loop occurs intramolecularly (cis) and phosphorylation of IN-box occurs intermolecularly (trans), we used the kinetic test developed by *Dodson et al., 2013*. To this end, we used different concentrations of enzyme complexes with fixed high substrate concentrations and continuously followed the product

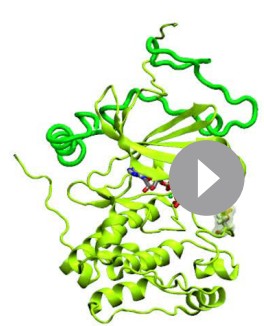

**Video 4.** 2.8 µs molecular dynamics (MD) simulation trajectory of [Aurora B/IN-box]^loop-P^.
https://elifesciences.org/articles/85328/figures#video4

formation (*Figure 5—figure supplement 2A*). The traces of product formation were normalized to the respective enzyme concentrations (*Figure 5E*). An overlap of the normalized traces in the lag phase of the curve indicates a concentration-independent, intramolecular, reaction. This profile was observed for [Aurora B/IN-box]$^{\text{IN-P}}$ (pink curves), confirming that phosphorylation of the activation loop occurs in 'cis'. In the case of [Aurora B/IN-box]$^{\text{loop-P}}$ (green curves), the normalized traces of product formation at different enzyme concentrations show that the rate of product formation is concentration dependent. At higher enzyme concentration, the changes in the slope are more pronounced, confirming that phosphorylation of the TSS motif occurs intermolecularly. Thus, our kinetic analysis with partially phosphorylated intermediates identifies autophosphorylation of the Aurora B activation loop as a cis-acting, rate-limiting step and IN-box TSS phosphorylation as a faster trans-acting event in the autoactivation of [Aurora B/IN-box].

It is noteworthy that phosphorylation of the activation loop in cis is the first necessary step in the autoactivation process, assuming a completely unphosphorylated enzyme pool. However, a partially active or fully active enzyme can phosphorylate the activation loop in trans. This type of activation mechanism with an initial intramolecular activation step followed by an intermolecular step of activation have been previously reported for PAK2 (*Wang et al., 2011b*) and for Aurora B (*Zaytsev et al., 2016*).

## The IN-box plays a crucial role in both organizing the substrate peptide binding site and enabling catalysis of the [Aurora B/IN-box] complex

According to the MD simulation the arginine residues of IN-box, IN-box$^{\text{Arg843}}$ and IN-box$^{\text{Arg847}}$, interact with Aurora B$^{\text{pThr248}}$ in the intermediate [Aurora B/IN-box]$^{\text{loop-P}}$ and the fully phosphorylated enzyme complex, respectively. To investigate the role of these arginine residues in kinase activity, we generated point mutants IN-box$^{\text{Arg843Ala}}$ and IN-box$^{\text{Arg847Ala}}$ in the context of the [Aurora B/IN-box] complex. Mutation of IN-box$^{\text{Arg847}}$, which interacts with Aurora B$^{\text{pThr248}}$ in the final structure of [Aurora B/IN-box]$^{\text{all-P}}$, to alanine results in an enzyme complex with four times lower catalytic efficiency ($v_{\text{max}}/K_M$) (*Table 1*). This is due to the fivefold increase in $K_M$ for the substrate peptide compared with the wt complex, while the $k_{\text{cat}}$ increases only slightly. Thus, kinetic analysis identifies this residue and its interaction with Aurora B$^{\text{pThr248}}$ as an important organizer of the substrate-peptide binding site. Interestingly, in the [Aurora C/IN-box] crystal structure this residue makes strong salt bridge with Aurora C$^{\text{Glu98}}$ (corresponding to Aurora B$^{\text{Glu148}}$) located in the Aurora B$^{\alpha C}$ helix. On the other hand, mutation of the IN-box$^{\text{Arg843}}$ to alanine results in 2.5-fold decrease in catalytic efficiency. This residue is involved in stabilizing Aurora B$^{\text{pThr248}}$ in [Aurora B/IN-box]$^{\text{loop-P}}$ and it is stabilizing the IN-box TS$^{\text{P}}$S$^{\text{P}}$ conformation in [Aurora B/IN-box]$^{\text{all-P}}$ simulations, as well as in the [Aurora C/IN-box] crystal structure, albeit the orientation of residues is different. The mutant IN-box$^{\text{Arg843Ala}}$ is not affected in the $K_M$ for the substrate peptide, suggesting that the IN-box$^{\text{Arg843}}$ predominantly supports the catalytic process (*Table 1*). The residue is far from the active site of the enzyme complex but could be involved in stabilizing the enzyme conformations and/or dynamics, favoring efficient catalysis. These results again suggest a dual role of the IN-box residues (which is partially uncoupled in the tested mutants) in both substrate recognition and catalysis.

## Maximal activation by phosphorylation is associated with concerted movement between the [Aurora B/IN-box] subdomains and the activation loop

Next, we analyzed the changes in global movements of the [Aurora B/IN-box] in MD simulations as a function of the phosphorylation status of the enzyme complex. We performed principal component analysis (PCA) for the MD trajectories of Aurora B in each of the phosphorylation states. The first principal components (eigenvectors) are always associated with the largest conformational changes, and this type of analysis has proven very useful for understanding global movements that govern catalysis in other protein kinases (*Masterson et al., 2011*). The dominant eigenvectors (*Figure 6—figure supplement 1A*) for all enzyme forms describe the opening and closing of the active site due to the two lobes of the kinase becoming distant or closer respectively, the other dominant motion is the twisting or shearing motion between the kinase lobes, as well as the movements in the activation loop (*Figure 6A*). However, the type of movement defined by the first principal component (PC1, the one corresponding to the largest relative movement) depends on the phosphorylation status of the enzyme complex. This is open-close movement (for [Aurora B/IN-box]$^{\text{no-P}}$), twisting (for [Aurora

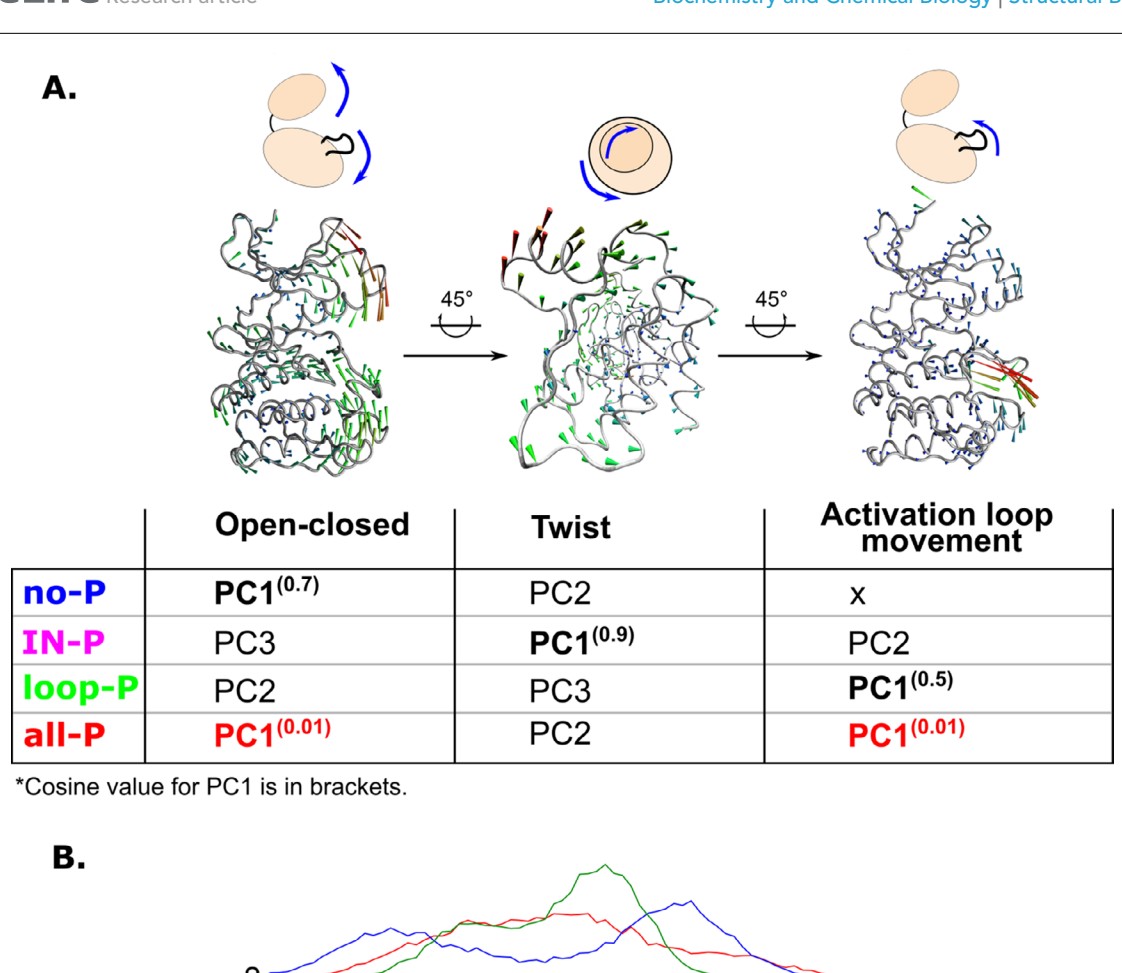

**A.**

| | Open-closed | Twist | Activation loop movement |
|---|---|---|---|
| **no-P** | **PC1$^{(0.7)}$** | PC2 | x |
| **IN-P** | PC3 | **PC1$^{(0.9)}$** | PC2 |
| **loop-P** | PC2 | PC3 | **PC1$^{(0.5)}$** |
| **all-P** | **PC1$^{(0.01)}$** | PC2 | **PC1$^{(0.01)}$** |

*Cosine value for PC1 is in brackets.

**B.**

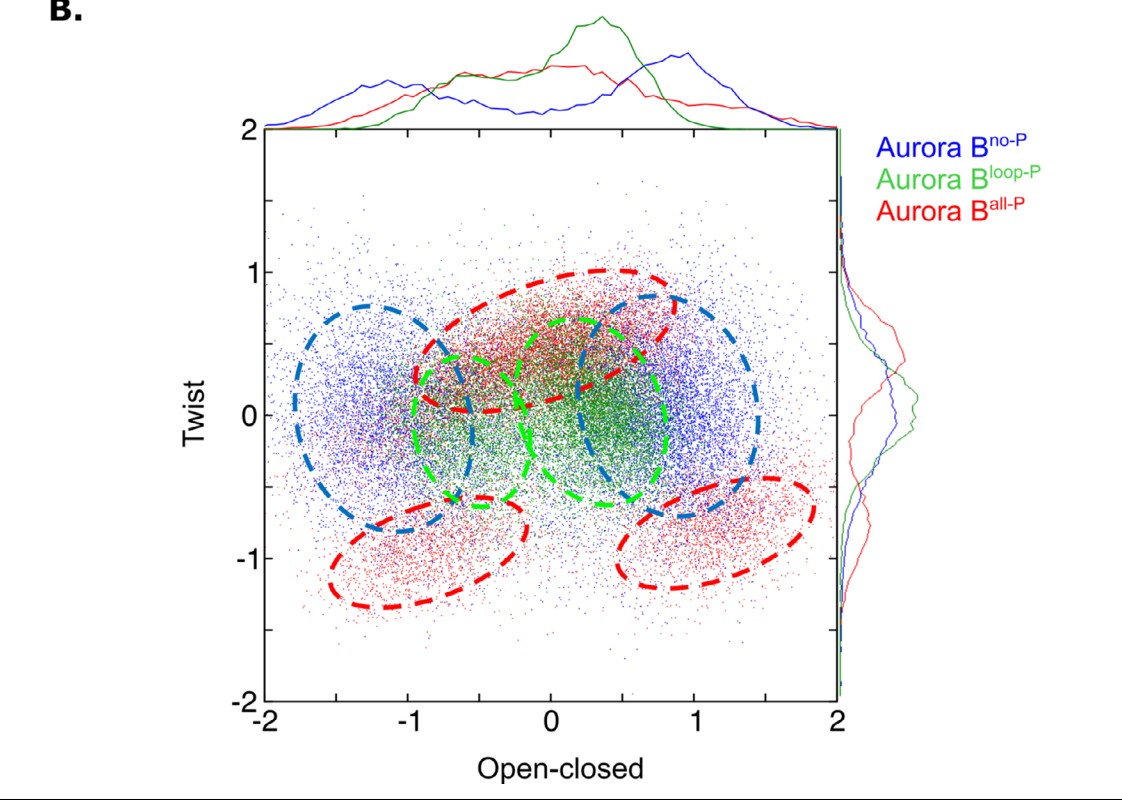

Aurora B$^{no-P}$
Aurora B$^{loop-P}$
Aurora B$^{all-P}$

**Figure 6.** Principal component analysis (PCA) shows that only fully phosphorylated [Aurora B/IN-box] achieves coordinated movements between opening/closing, twisting, and movements in the activation loop. (**A**) Protein movements in the different phosphorylation forms of the [Aurora B/IN-box] complex in PC1 are illustrated by porcupine diagrams (opening/closing movement, twisting movement, movement in the activation loop). The needle tip indicates the direction of the vector and the magnitude, while the color (blue-green-red) corresponds to the increasing magnitude. The table

*Figure 6 continued on next page*

*Figure 6 continued*

summarizes the results of PCA for different phosphorylation forms of the complex. The cosine value close to 0, indicating an ordered movement, was obtained only for PC1 of [Aurora B/IN-box]$^{all-P}$, implying a coordinated movement of opening and closing and a restructuring of the activation loop. (**B**) Correlation between 'opening/closing' and 'twist' for [Aurora B/IN-box]$^{no-P}$ (blue), [Aurora B/IN-box]$^{loop-P}$ (green), and [Aurora B/IN-box]$^{all-P}$ (red).

The online version of this article includes the following figure supplement(s) for figure 6:

**Figure supplement 1.** Principal components MD data analysis.

B/IN-box]$^{IN-P}$), an activation loop movement (for [Aurora B/IN-box]$^{loop-P}$), and synchronized opening and closing of lobes and loop movement (for [Aurora B/IN-box]$^{all-P}$) (*Video 5*). Importantly, the cosine content evaluation (*Hess, 2000*) for PC1 in [Aurora B/IN-box]$^{all-P}$ was only 0.01. In contrast, the first eigenvectors in all other forms had a cosine content between 0.5 and 0.9 (*Figure 6A*, table). The small values of the cosine content indicate that the deformation in [Aurora B/IN-box]$^{all-P}$ is a truly organized vibrational deformation that coordinates the open-closed/twist motion with the oscillation of the activation loop, whereas the first mode in all other phosphorylation states of the enzyme complex (with high cosine content) is mostly associated with random deformations.

We then analyzed the relationship between the open-closed movement and the twist movement (*Figure 6B*). The plot shows that the unphosphorylated form of the enzyme complex, [Aurora B/IN-box]$^{no-P}$ (blue), explores a wider range of conformations, consistent with a highly disorganized, easily deformable structure. Along the 'open-closed' axis, this enzyme complex forms two groups of ensembles, clearly indicating the existence of two stabilized populations (one in the open and the other in the closed state; blue circles in the *Figure 6A*), while intermediate conformations are sparse. Phosphorylation of the activation loop, [Aurora B/IN-box]$^{loop-P}$ (green), significantly reduces the conformational space examined along both vectors. This reflects a better organized enzyme with closer distance between the lobes, consistent with progressive restriction of movement within the structure. Finally, full phosphorylation, [Aurora B/IN-box]$^{all-P}$ (red), further increases the degree of order and rigidity within the enzymatic complex, forcing the enzyme into three well-defined regions in the graph that correspond to the occurrence of natural oscillations associated with opening and closing movements and twist (*Figure 6—figure supplement 1B–C*). Negative twisting is associated with two distinct conformational basins, corresponding to the closed or open conformation of the enzyme. The separation of the two basins indicates that an open-closed transition is unlikely in this twist conformation. On the contrary, when the twist between the lobes is moderately positive, the enzyme can easily transition between the open and closed states. It can be concluded that in [Aurora B/IN-box]$^{all-P}$, the transitions between open and closed conformation are regulated and coordinated by the twist, indicating complete synchronization of the global movements of the kinase subdomains when fully phosphorylated.

In summary, only when the enzyme complex is fully phosphorylated are the opening and closing of the lobes, the shearing motion between the lobes, and the opening and closing of the activation loop fully coordinated, resulting in efficient catalysis. The PCA result is consistent with the kinetic and HDX analysis and highlights that full activity requires a well-structured enzyme complex with a high degree of coordination of movement, which is synergistically achieved by phosphorylation in the Aurora B activation loop and in the IN-box.

## Discussion

Using HDX and MD, we found that phosphorylation has a structuring effect on the [Aurora B/IN-box] complex, with the enzyme exploring fewer conformations and therefore becoming more rigid. Structuring after phosphorylation appears to be a general activation mechanism for

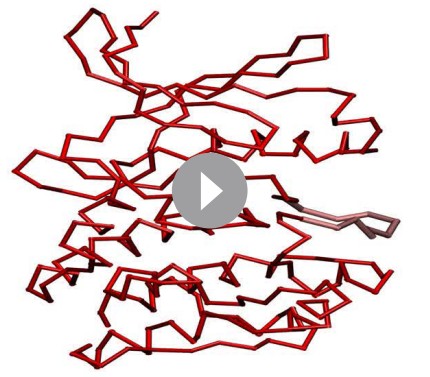

**Video 5.** PC1 for [Aurora B/IN-box]$^{all-P}$: coordinated motions of open-closed/twist and activation loop movements.

https://elifesciences.org/articles/85328/figures#video5

the protein kinase family, and similar effects have been reported for protein kinase A and others (*Hsu et al., 2008*; *Steichen et al., 2010*).

A unique feature of Aurora B is its obligate association with the binding partner INCENP, which is also subject to phosphorylation. The recent crystal structure of [Aurora C/IN-box] showed that the phosphorylated IN-box interacts with the phosphorylated activation loop and stabilizes a more compact conformation, compatible with the active state of kinases. This structure implies a synergistic interaction between the two phosphorylation sites of the enzymatic complex (*Abdul Azeez et al., 2019*). We complement and extend these findings by using an experimental approach that is independent of the ability to crystallize the protein complex. Our approach addresses the lack of structural information for the unphosphorylated complex and the lack of information about structural dynamics and its possible role in [Aurora B/IN-box] autoactivation. Using a solution-based and computational approaches, we methodically studied the unphosphorylated, phosphorylated, and partially phosphorylated forms of the enzyme complex under the same conditions. Our enzyme kinetics experiments with the partially phosphorylated forms confirm that the two phosphorylation sites contribute synergistically to the Aurora B kinase activity. Moreover, the HDX experiments together with the MD simulations show a correlation between the dynamics of the two partners (Aurora B and the IN-box) and that phosphorylation of the activation loop in conjunction with phosphorylation in the IN-box is required to coordinate the dynamics between the N-terminal and C-terminal lobes of the kinase.

The HDX experiment clearly shows that the IN-box is a highly dynamic polypeptide chain, with most of its regions reaching maximal deuteration after only 3 min of $D_2O$ incubation (*Supplementary file 5*), even in the phosphorylated state. Interestingly, the N-terminal part of the IN-box, which is the only part containing defined secondary structure elements, exhibits EX1-like HDX kinetics, suggesting that the IN-box$^{\alpha A}$ and IN-box$^{\alpha B}$ helices reversibly undergo folding and unfolding. Despite the reversible unfolding of the helices, the IN-box remains associated with Aurora B, which was confirmed by our mass photometry experiments (*Figure 2* and *Figure 1—figure supplement 4*). The EX1 kinetics, and its relation to local unfolding of proteins under native conditions, has been described in detail in

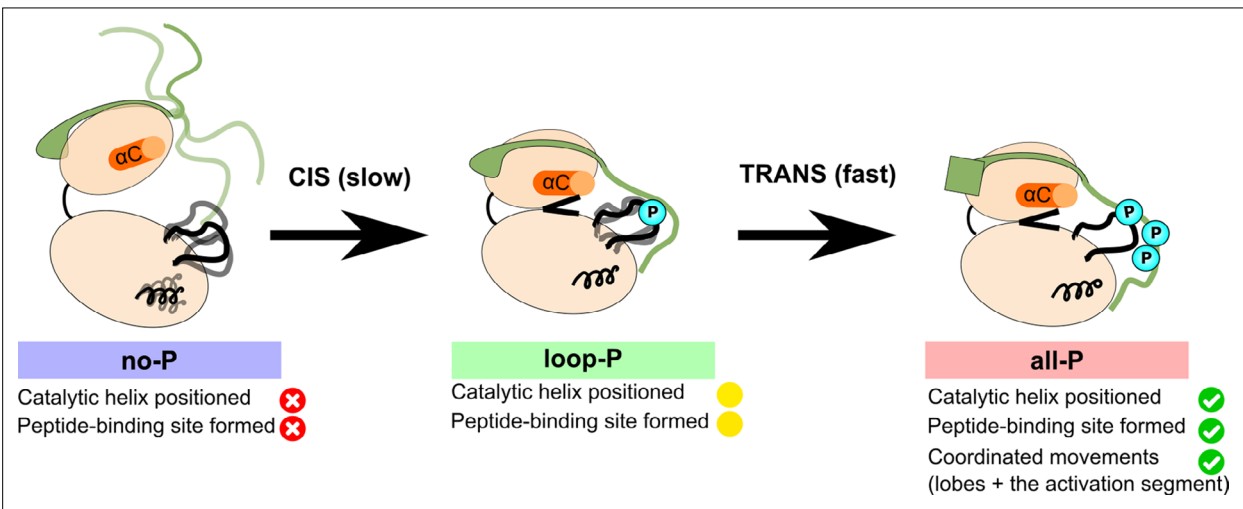

**Figure 7.** Model summarizing the autoactivation mechanism of [Aurora B/IN-box]. In the absence of phosphorylation, the [Aurora B/IN-box]$^{no-P}$, the enzyme complex is very flexible. The activation segment and Aurora B$^{\alpha G}$ are disorganized, Aurora B$^{\alpha C}$ is not in catalytic position and the two lobes of the kinase are more separated. The IN-box is disorganized and its C-terminal region does not interact with Aurora B. This enzyme can autophosphorylate its own activation loop in a slow intramolecular process. Once the activation loop is phosphorylated, [Aurora B/IN-box]$^{loop-P}$, Aurora B$^{\alpha G}$ is stabilized and the IN-box C-terminal region is closer to the C-lobe of Aurora B. This leads to the stabilization of Aurora B$^{\alpha C}$ and partial formation of the substrate binding site around the phosphorylated activation loop. This intermediate is further phosphorylated in the IN-box in a faster intermolecular process. In the fully phosphorylated form of the enzyme complex, [Aurora B/IN-box]$^{all-P}$, the IN-box stably interacts with the phosphorylated loop; Aurora B$^{\alpha C}$ is in the productive position and the peptide binding site is fully formed. The enzyme complex now has all functional elements in place and synchronization of key global motions of the kinase lobes ensures efficient catalysis.

The online version of this article includes the following figure supplement(s) for figure 7:

**Figure supplement 1.** Root mean square fluctuations (RMSFs) for IN-box chain in the context of [Aurora C/IN-box]$^{all-P}$ (magenta) are more stable than in the [Aurora C/IN-box]$^{loop-P}$ (cyan).

the literature. (*Arrington and Robertson, 2000*; *Chitta et al., 2009*; *Ferraro et al., 2004*; *Krishna et al., 2004*; *Skinner et al., 2012*; *Sperry et al., 2012*; *Walters et al., 2013*).

We observed EX1 kinetics in the N-terminal part of the IN-box independent of phosphorylation status, but the EX1 kinetics are faster in the unphosphorylated state, suggesting that the IN-box[805-825] fragment in [Aurora B/IN-box][no-P] switches more rapidly between the folded and unfolded states. This observation, together with the MD simulation analysis indicates that the higher unfolding rate in the IN-box N-terminal region of the unphosphorylated state transfers disorder to the adjacent N-lobe of Aurora B and in this way negatively affects kinase activity (*Figure 2D and F*). In summary, we provide strong evidence that in the unphosphorylated form of the enzyme complex, the IN-box acts as a negative regulator.

On the other hand, phosphorylation of the Aurora B activation loop leads to partial structuring within Aurora B and to a substantial conformational change of the IN-box, as well as to the adoption of an enzyme conformation closer to that required for efficient catalysis. However, the absence of phosphate groups on the IN-box continues to interfere with the optimal arrangement of the enzyme complex. Finally, when both Aurora B and the IN-box are phosphorylated, the enzyme complex is structured and coordinated in its global movements (*Figure 7*). Thus, in the fully phosphorylated form of the enzyme complex, the phosphorylated IN-box acts as a positive regulator that stabilizes the productive enzyme conformation.

The results of the MD simulations are consistent with the HDX analysis and highlight the entropic nature of the IN-box. However, the C-terminal region of the IN-box adopts a different conformation in our [Aurora B/IN-box][all-P] simulation than in the [Aurora C/IN-box][all-P] crystal structure, raising the possibility that the observed [Aurora B/IN-box][all-P] structure is either an intermediate on the activation coordinate or an alternative conformation in the fully phosphorylated state. To better understand the role of phosphorylation on the IN-box conformation, we performed simulations on [Aurora C/IN-box] in the fully phosphorylated state, [Aurora C/IN-box][all-P], and in the partially phosphorylated state, [Aurora C/IN-box][loop-P], using the [Aurora C/IN-box] crystal structure as a starting model (*Figure 7—figure supplement 1*). These simulations confirmed that the residues following the TSS motif are highly flexible, even in the fully phosphorylated state. In addition, MD simulations on an [Aurora C/IN-box][loop-P] show that the absence of the IN-box phosphates in the [Aurora C/IN-box] structure results in relaxation of the longer C-terminal segment of the IN-box, which contains the TSS motif. In this simulation, the IN-box is less organized, overall (higher RMSD), than in the [Aurora C/IN-box][all-P] simulation and the IN-box[886-903] rapidly loses association with the phosphorylated activation loop and moves toward the C-lobe of the kinase to adopt a similar, but not identical, conformation to that observed in our original simulation, [Aurora B/IN-box][loop-P] (*Figure 7—figure supplement 1*, inset). The observed behavior confirms that the C-terminal part of the IN-box is a highly entropic unit that follows fast diffusion dynamics directly controlled by the phosphorylation state of the system, with the highest degree of disorder in the unphosphorylated state, partially organized in the [Aurora B(C)/IN-box][loop-P] state, and better organized, although still highly dynamic, in the [Aurora B(C)/IN-box][all-P].

The high dynamics of the IN-box is most likely the main reason for the difficulties in obtaining the crystal structure of [Aurora B/IN-box][all-P], as reported by *Sessa et al., 2005*. Therefore, according to our analysis, it is likely that there are few equally stable and productive conformations of [Aurora B/IN-box][all-P]. [Aurora C/IN-box][all-P] more likely represents the more stable conformation of the active state while the conformation observed in the [Aurora B/IN-box][all-P] simulation, described here, is probably a productive intermediate along the activation coordinate. Limitations in the simulation time and the very entropic nature of the IN-box C-terminal region may be the reasons our simulation did not reach the IN-box conformation observed in the [Aurora C/IN-box][all-P] crystal structure.

Overall, the IN-box plays a major role in the structural conformations adopted by Aurora B and in the synchronization of Aurora B subdomain motions observed in the active state of the enzyme complex. The domain movements and correlated motions within kinase lobes have been proposed to be part of the enzymatic turnover (*Kumar et al., 2018*; *Kuzmanic et al., 2017*; *Masterson et al., 2010*; *Xiao et al., 2014*). The strongest evidence for the coupling of global dynamics of protein kinase lobes with catalysis comes from NMR experiments on protein kinase A, where the opening and closing of the active site correlates very well with the catalytic turnover of the enzyme (*Masterson et al., 2011*). Since only the fully active [Aurora B/IN-box][all-P] exhibits synchronized global motions, we

propose that global domain movement is directly linked to the catalytic cycle of this enzyme complex and likely controls substrate binding and product release.

In the present study, we use the [Aurora B/IN-box] complex containing only the INCENP fragment required for interaction with Aurora B, the IN-box, whereas the full-length INCENP has a long N-terminal part, which raises the question of whether or not the dynamics of the IN-box are different in the context of the full-length INCENP. The IN-box was previously defined as the minimal region of INCENP that binds Aurora B (*Sessa et al., 2005*), and phosphorylation of [Aurora B/IN-box] results in similar enzymatic activation as [Aurora B/INCENP^full-length] (*Wang et al., 2011a*; *Zaytsev et al., 2016*), suggesting that the IN-box is the only region of INCENP that is important and necessary for the regulation of kinase activity by phosphorylation. This is a strong indication that the dynamic changes observed in the IN-box upon phosphorylation of [Aurora B/IN-box] most likely follow the same trend in the context of full-length INCENP. At the same time, it is noteworthy that the N-terminal region of INCENP is required for the localization of the CPC at the inner centromere, resulting in a high local concentration of the [Aurora B/INCENP] complex, which is crucial for the rapid intermolecular activation of the kinase complex (*Wang et al., 2011a*; *Zaytsev et al., 2016*).

It has been previously proposed that autoactivation of [Aurora B/IN-box] occurs via a two-step mechanism with a slow cis-step and a fast trans-step (*Zaytsev et al., 2016*). Following our kinetic and MD analysis with partially phosphorylated enzyme complexes, we can now confirm this model and conclude that phosphorylation of the Aurora B activation loop is the slow intramolecular process followed by rapid trans phosphorylation of the IN-box TSS motif. It is important to emphasize that in the proposed model, the slow intramolecular step is obligatory only when the entire population of [Aurora B/IN-box] is dephosphorylated. The initial intramolecular step is followed by intermolecular propagation of activation, which may also include the activation segment.

Phosphorylation gradients of Aurora B substrates in the cell have been identified (*Fuller et al., 2008*; *Wang et al., 2011a*), and these long-range gradients play a regulatory role in mitosis (*Afonso et al., 2014*; *Ferreira et al., 2013*; *Uehara et al., 2013*). In addition, a strong evidence for the existence of a steep gradient of Aurora B activity at metaphase chromosomes has been reported and it has been suggested that the kinetics of Aurora B activity plays an important role in the formation of this gradient (*Liu et al., 2009*; *Zaytsev et al., 2016*). Here, we found that the partially phosphorylated [Aurora B/IN-box] intermediates have very low kinase activity (5–10%) which, together with the two-step Aurora B autoactivation, is consistent with the ability to form a steep phosphorylation gradient. Thus, a small increase in phosphatase concentration in a local cell region such as the kinetochore could cause a sharp decrease in kinase activity in the kinetochore region, whereas in a nearby region, Aurora B kinase activity could remain high.

The very similar Aurora A (72% identity), which associates with a different binding partner, TPX2, can be activated to an appreciable extent independently, either only by association with TPX2 or only by phosphorylation in the activation loop (*Dodson and Bayliss, 2012*). Two pathways of Aurora A activation are used separately at the physiological level. The pool of Aurora A involved in chromatin-driven microtubule nucleation is regulated exclusively by association with TPX2 (*Kufer et al., 2002*) and phosphorylation in the activation loop is not observed (*Levinson, 2018*). On the other hand, Aurora A activation at the centrosome depends only on phosphorylation of the activation loop and does not involve TPX2 binding (*Joukov et al., 2010*). While Aurora A has evolved to allow two independent activation pathways, activation of Aurora B, which is always in complex with IN-box, follows a single synergistic activation pathway that requires the coincidence of two events: phosphorylation of threonine in the activation loop and phosphorylation of the binding partner (TSS motif in the IN-box).

The present study demonstrates the important role that dynamics plays in regulating the enzymatic activity of [Aurora B/IN-box]. Our study explains and highlights the importance of IN-box (specific and obligatory Aurora B partner) as a necessary structural and strong regulatory element of the enzyme complex. The new findings suggest that the interface between Aurora B and the IN-box represents a target area for the development of a new type of Aurora B-specific drugs that would disrupt the allosteric changes required for activation. Our results contribute to a better understanding of the biophysical basis for the switch-like transition in Aurora B catalytic activity upon autophosphorylation (*Zaytsev et al., 2016*), a feature that underlies the intracellular spatial pattern of Aurora B substrate phosphorylation essential for proper chromosome segregation and cell division.

## Materials and methods

### Preparation of [Aurora B/IN-box] complex in different phosphorylation states

The wt enzymatic complex containing Aurora B (60–361 aa) and IN-box (790–856 aa) from *X. laevis* was expressed in *Escherichia coli*, using a bicistronic construct that was cloned in the pRSF duet vector (Novagen). The GB1 domain from the streptococcal protein G was fused to the N-terminus of Aurora B in order to increase protein expression and solubility. BL21(DE3)pLysS cells were transformed with the mentioned plasmid and protein expression was induced with 0.2 mM of IPTG at OD of 0.6 and incubated for 18 hr at 18°C with shaking at 180 rpm.

Protein purification was conducted in two steps. First, IMAC with nickel column (GE HealthCare) was used to trap the tagged protein complex. After removal of the 6XHisGB1 by TEV cleavage, the enzyme complex was further purified with size exclusion chromatography. The protein was stored at –80°C in a buffer containing 10 mM HEPES pH 7.5, 150 mM NaCl, 2.5 mM DTT, and 50% glycerol until further use. To obtain the dephosphorylated form, the enzyme complex was incubated with 0.2 µM of GST-$\lambda$ phage phosphatase (made in-house) during 1 hr at 30°C, followed by incubation with gluta-thione beads to remove the phosphatase.

To generate the partially phosphorylated forms of the enzyme, the 6XHis-GB1-Aurora B (59–361 aa) and IN-box (790–844 aa; IN-box$^{\Delta C}$) coding regions were introduced in the pTXB1 plasmid (NEB) by in-fusion cloning using the Nde1 and Sap1 cloning site. The resulting construct has two transla-tional cassettes. The upstream cassette contains 6xHis-GB1-(TEV site)-Aurora B and the downstream cassette contains the IN-box$^{\Delta C}$ fused to the N-terminus of the Mxe intein-chitin binding domain protein (*Telenti et al., 1997*). The enzyme complex, [Aurora B/IN-box]$^{IN-\Delta C}$, was expressed using the same parameters as the wt. Furthermore, IMAC was used as the first step of purification. Next, for obtaining the [Aurora B/IN-box]$^{loop-P}$, the enzyme complex was incubated with the fully active [Aurora B/IN-box] (10 nM final) in the presence of 5 mM ATP-Mg for 1 hr. This ensured full phosphorylation of Aurora B$^{Thr248}$. Analogously, for obtaining [Aurora B/IN-box]$^{IN-P}$, the enzyme complex was incubated with GST-$\lambda$-phosphatase (0.2 µM final) for 1 hr. This ensured full dephosphorylation of Aurora B$^{Thr248}$. In parallel with either phosphorylation or dephosphorylation of the enzyme complex, the reaction was incubated with TEV protease for 3 hr to remove the 6xHis-GB1 tag. The enzyme complex was then further purified by incubation with chitin beads (NEB), to remove traces of TEV and fully active kinase or phosphatase. Elution from the chitin beads was done by overnight incubation with 50 mM of 2-mercaptoethanesulfonic acid (MESNA) (Sigma-Aldrich) that induces intein-mediated cleavage. The product is a C-terminal thioester on the IN-box$^{\Delta C}$ which can be ligated to a synthetic peptide with an N-terminal cysteine. At this step, in order to obtain [Aurora B/IN-box]$^{loop-P}$, we fully incubated phos-phorylated [Aurora B/IN-box]$^{IN-\Delta C}$ with the peptide sequence CKRTSSAVWHSPPL-6xHis. Analogously, in order to obtain [Aurora B/IN-box]$^{IN-P}$ we incubated dephosphorylated [Aurora B/IN-box]$^{IN-\Delta C}$ with the peptide sequence CKRTS(PO$_4$)S(PO$_4$)AVWHSPPL-6xHis. The ligation reaction proceeded over-night in a ligation buffer (50 mM HEPES pH 8.0, 300 mM NaCl, 10 mM MESNA, 5 mM ascorbic acid, 10 mM TCEP) with 50 µM [Aurora B/IN-box]$^{\Delta C}$ and 600 µM peptide. To remove the excess of unligated peptide, the protein sample was run on Superdex S200 16/60 column. To separate the ligated enzy-matic complex from the unligated [Aurora B/IN-box]$^{IN-\Delta C}$, the sample was run over a Ni column and the pure ligated protein was eluted with imidazole. The protein was stored same as wt until further use.

### Enzymatic assay

The enzymatic activity of Aurora B was measured by monitoring the product (ADP) formation rate using a pyruvate kinase/lactate dehydrogenase-coupled spectrophotometric assay (*Roskoski, 1983*; *Wu and Wang, 2003*). The peptide substrate used was ALRRFSLHGA, which was acetylated at the N-terminus and amidated at the C-terminus. Kinetic measurements were performed in kinase assay buffer (25 mM Tris-HCl pH 7.4, 100 mM KCl, 2 mM MgCl$_2$, 1 mM EGTA, 2 mM DTT, 1 mM ATP, 160 µM NADH, 0.5 mM phosphoenol pyruvate containing 20 U/ml lactate dehydrogenase and 20 U/ml pyru-vate kinase) with substrate peptide at the concentrations indicated in the experiment. The reaction was initiated by addition of the [Aurora B/IN-box] complex, and the progress of the reaction was continuously monitored by tracking the decrease in absorbance at 340 nm using a Cary UV 60 spec-trophotometer (Agilent). The initial rate was determined by calculating the slope of product release in

the linear portion of the reaction, which occurs in the first few minutes of the reaction when less than 10% of the substrate has been converted to products.

## Aurora B autoactivation assay

The autoactivation of [Aurora B/IN-box]$^{no-P}$, [Aurora B/IN-box]$^{loop-P}$, or [Aurora B/IN-box]$^{IN-P}$ was assessed by preincubating 12 µM of the enzymatic complex in a buffer containing 20 mM Tris pH 7.4, 100 mM KCl, and 5 mM ATP-Mg. The autophosphorylation reaction was left to proceed and aliquots were taken at different times of preincubation with ATP to determine the initial rates. Each aliquot was diluted to a low nanomolar concentration (10–50 nM in the kinase assay buffer supplemented with 0.6 mM substrate peptide) and initial rates were determined as described above. Since low nanomolar concentrations of Aurora B were used in the assay and the rate is determined in the first minutes of the reaction (2–4 min, where the kinetics of product release are linear), the amount of autophosphorylation of the enzyme is negligible.

## Kinetic test to determine cis or trans activation steps

In difference to the autoactivation assay, where enzyme complex was preincubated with ATP-Mg, here the enzyme complexes at different concentrations (25, 50, and 100 nM) were incubated with 1 mM ATP-Mg and 0.6 mM substrate peptide at the same time, and the formation of ADP was continuously monitored for a prolonged period (450 min). It should be noted that the traces of product formation are not linear due to the autoactivation that is happening during the experiment. The traces of ADP formation are normalized to the concentration of the enzyme complex to evaluate the concentration effect on the autoactivation process, as reported by *Dodson and Bayliss, 2012*. Overlapping curves in the lag phase of the product formation indicate concentration-independent activation (cis) and curves with different slopes indicate concentration-dependent activation (trans).

## HDX assay

The on-exchange reaction was done by diluting [Aurora B/IN-box] enzymatic complex 1:20 in a the on-exchange buffer (20 mM HEPES pD 7.0, 300 mM NaCl, 2 mM DTT, 1 mM ADP, and 2 mM MgCl$_2$, in D$_2$O) so that the final D$_2$O concentration was 95%. The on-exchange reaction was quenched at different time points (3, 10, 30, 90, and 300 min) by mixing 50 µl of the reaction with equal volume of ice-cold quench buffer (250 mM potassium phosphate pD 2.3) and flash freezing in liquid nitrogen. The quenched samples were stored at –80°C before proteolysis and liquid chromatography-mass spectrometry steps. The HDX experiment was performed three times independently (biological replicates) for the wt [Aurora B/IN-box]$^{all-P}$ and [Aurora B/IN-box]$^{no-P}$ and with one biological replicate for the intermediates phosphorylated forms. All the biological replicates show consistent results. The reported data for all forms is for a single biological replicate where each time point was run two or three times (technical replicates).

## Liquid chromatography and MS

The quenched samples were thawed on ice and injected on a nanoACQUITY UPLC system with HDX technology (Waters). The temperature of the chamber containing the sample loop as well as the UPLC and trap columns was set at 0.5°C. The temperature of the pepsin column compartment was set at 10°C. The quenched samples (10 pmol) were injected into a 50 µl sample loop and run in a trapping mode, where the protein was passed through a pepsin column (Waters Enzymate 2.1×30 mm, 5 µm) and the proteolyzed sample was immediately directed to a trap column (Waters Acquity Vanguard BEH C18, 1.7 µm, 2.1×5 mm) to desalt peptides fragments. The flow rate was set to 70 µl/min during the first minute, followed by 100 µl/min for another 2 min with buffer A (0.2% formic acid, 0.01% trifluoroacetic acid pH 2.5). After desalting, the peptides were separated by C18 analytical column (Waters Acquity BEH C18, 1.7 µm, 1.0×100 mm) with a linear 5–50% acetonitrile gradient using buffer B (99.9% acetonitrile, 0.1% formic acid, and 0.01% trifluoroacetic acid pH 2.5). The elution gradient was run at 40 µl/ min for 17 min. The output of the analytical column was directed to a mass spectrometer (Q-TOF SYNAPT G2-Si, Waters) for peptide identification and determination of the deuterium uptake. The mass spectrometer was operated in the positive ion electrospray mode, with the ion mobility function to minimize spectral overlap using the MS$^E$ acquisition mode (Waters Corporation). Lock mass correction with the Leu-ENK peptide was used to ensure mass accuracy determination.

To prepare fully deuterated sample, the unlabeled enzyme complex was injected to the HPLC system for digestion on pepsin column under the same condition as other samples. However, the peptides eluted from the analytical column were collected in one fraction instead of being injected to the MS. The peptides were then lyophilized and resuspended in the on-exchange buffer and the reaction was injected into the HPLC system connected to MS in the same way it was for the other samples but bypassing the pepsin column.

## HDX-MS data evaluation

A library of non-deuterated peptides was created using the ProteinLynx Global server 3.0 (PLGS) (Waters) using the following requirements: (1) a mass error for the peptide has to be below 10 ppm for the precursor ion, (2) the peptide has to have at least two fragmentation products, and (3) the peptide has to be identified in at least two out of three non-deuterated runs. The level of deuteration in the peptides was determined with DynamX 3.0 (Waters). A manual inspection of all the assignment was conducted to confirm the data or discard the noisy or overlapping spectra.

The difference in deuteration between two states (ΔD= [Aurora-B/IN-box]$^{no-P}$ - [Aurora-B/IN-box]$^{all-P}$) was expressed in percentage and was calculated by normalization with respect to the theoretical maximum uptake (MaxUptake). The maximum uptake is determined as follows: MaxUptake =NP-2, where N is the number of amino acids in the peptide and P is the number of prolines. The percentage of deuteration was determined according to formula: $\Delta D\left(\%\right) = \frac{D}{\text{Max uptake}} * 100$.

For the partially phosphorylated constructs ([Aurora-B/IN-box]$^{loop-P}$ and [Aurora-B/IN-box]$^{IN-P}$) the difference in deuteration was determined with respect to [[Aurora-B/IN-box]$^{no-P*}$], for example for [Aurora-B/IN-box]$^{loop-P}$ the difference in deuteration is (ΔD= [Aurora-B/IN-box]$^{no-P*}$- [Aurora-B/IN-box]$^{loop-P}$). The ΔD values were normalized with respect to the fully deuterated peptides and expressed in percentage. $\Delta D\left(\%\right) = \frac{D}{FD} * 100$, where FD is the amount of deuterium experimentally determined in the fully deuterated sample. The [Aurora-B/IN-box]$^{all-P*}$ and [Aurora-B/IN-box]$^{no-P*}$ constructs contain the mutation IN-box$^{Phe845Cys}$ to correspond to the partially phosphorylated intermediates where this mutation is present due to the protein ligation process.

All peptides with a deuteration difference greater than 5% were considered different. Confidence intervals of 95% confidence for differential HDX-MS (ΔHDX) measurements of any individual time point were determined according to *Houde et al., 2011*, using the Deuteros software 2.0 (*Lau et al., 2021*).

To allow access to the HDX data of this study, the HDX data summary tables (*Supplementary files 1 and 3*) and the HDX data table (*Supplementary files 2 and 4*) are included in the supporting information as per consensus guidelines (*Masson et al., 2019*) and the raw data of the experiments were deposited in the PRIDE partner repository with the dataset identifier PXD038935.

## Mass photometry assay

The protein sample [Aurora-B/IN-box]$^{all-P}$ and [Aurora-B/IN-box]$^{no-P}$ were kept at 100 nM concentration in buffer containing 300 mM NaCl, HEPES 25 mM pH 7.5, and 2 mM DTT. Immediately before the measurement the sample was diluted to 10 nM. The data was acquired with Refeyn Two MP and analyzed with the Refeyn Discover MP software.

## MD simulations

We modeled the [Aurora B/IN-box] complex in four distinct phosphorylation states (no-P, loop-P, IN-P, all-P). Every model was independently built using the X-ray structure of the [Aurora B/IN-box]$^{loop-P}$ complex from *X. laevis* (PDB ID:4C2W; *Sessa and Villa, 2014*). The non-hydrolysable adenylyl-imidodiphosphate ATP mimic present in the original crystal structure was replaced with the ATP-Mg$^{2+}$ moiety taken from Aurora A structure (PDB ID:5DN3; *Janeček et al., 2016*). The $^{848}$TSSAVWHSP$^{856}$ C-terminal region of IN-box, not visible in the original crystal structure, was modeled using MODELLER (https://www.salilab.org/modeller/) as a random coil. The initial structures of the phosphorylation states for which an experimental structure is not available were modeled by introducing the pertinent phosphorylated groups where needed or removing the one in Aurora B$^{Thr248}$. The starting structures were completed by the addition of the missing hydrogens, choosing standard protonation at neutral pH for all titratable groups. Each of the four systems were solvated by around 36,000 water molecules and contained in a periodic box with an edge of 105 Å. The systems were completed by the addition

of 113 Cl⁻ and the corresponding Na⁺ required to achieve charge neutrality, resulting in a salt concentration close to the physiological value of 150 mM.

The CHARMM36 force field (*Huang et al., 2017*) was used to parameterize the protein segments, ATP, and the ions, while water was represented with the triangular TIP3P model (*Jorgensen et al., 1983*). All bonds involving hydrogen atoms were constrained to their equilibrium distances using the LINCS algorithm (*Hess et al., 1997*), the Lennard-Jones potentials were computed using a cut-off distance of 10 Å, while the electrostatic interactions were evaluated using the particle mesh Ewald method (*Essmann et al., 1995*).

The four systems were first relaxed to a low energy state by the steepest-descent algorithm until reaching a tolerance value of 1000 kJ/mol/nm on all forces. Then, the system was by MD in the NPT ensemble. The equations of motion were integrated using the leap-frog algorithm (*Hockney and Eastwood, 1988*) with a timestep of 2 fs. Pressure coupling was obtained using a Berendsen barostat, with coupling constant of 2 ps, and target pressure 1 Bar. Temperature equilibration was obtained by initially randomly setting the velocities of all atoms using a Maxwell-Boltzmann distribution at 10 K, and then by performing simulated annealing to 300 K within 2 ns of simulations. After equilibration, each system underwent a production run of 2.8 µs in the NPT ensemble, using the Nosé-Hoover chain thermostat (*Martyna et al., 1992*; *Nosé, 1984*; *Parrinello and Rahman, 1981*) with a coupling constant of 1 and 2 ps, respectively.

Simulations of [Aurora C/IN-box] were prepared and run in the same manner as those for [Aurora C/IN-box]. [Aurora C/IN-box]$^{all-P}$ was run for 1 µs and [Aurora C/IN-box]$^{loop-P}$ for 300 ns.

All simulations were performed and partly analyzed using the GROMACS package (*Pronk et al., 2013*; *Van Der Spoel et al., 2005*). Additional analysis and visualization were done using VMD (*Humphrey et al., 1996*).

Phosphorylation, as one of the most important post-translational modifications, has been thoroughly investigated by computational modeling over the years, in particular with great success for the kinase family (*Saladino and Gervasio, 2016*). Nonetheless, the involvement of strongly charged, titratable moieties challenge global transferability for the parameters, opening the ground to future improvement based on more sophisticated treatment of electrostatic interactions, like by polarizable force fields.

## Acknowledgements

The authors acknowledge the support of the Research Council of Norway through the Centre for Molecular Medicine Norway (Project No. 187615; NS, HG, OH, SMHW, and DSP), CoE Hylleraas Centre for Quantum Molecular Sciences (Project No. 262695; OH, MaC, and MiC), the Norwegian Supercomputing Program (NOTUR) (Project No. NN4654K; OH and MC) and individual Research Council of Norway grant (Project No. 325528) to NS. The work in the Black lab was supported by the NIH: Grant No. R35-GM130302 to BEB JMD-M was supported by NIH postdoctoral fellowship GM108360. We would also like to thank Prof. Magnus Kjærgaard (Århus University, Denmark) for performing the mass photometry measurements.

DSP, NS and BEB conceived the project. DSP carried out cloning, protein expression and purification, peptide design and ligation, kinetic characterization, HDX experiments, and HDX data processing and analysis. OH, MaC performed MD simulations under supervision of MiC OH, MaC, and MiC analyzed and interpreted MD data. JMD-M performed initial HDX experiments, designed and purified proteins, and performed initial phosphopeptide identification and quantitation under the direction of BEB HG and SMHW helped with protein purification. DSP and NS supervised the project and wrote the manuscript with input from all authors.

The authors declare that they have no conflict of interest.

## Additional information

### Funding

| Funder | Grant reference number | Author |
|---|---|---|
| Norges Forskningsråd | 187615 | Dario Segura-Peña<br>Oda Hovet<br>Hemanga Gogoi<br>Stine Malene Hansen Wøien<br>Nikolina Sekulic |
| Norges Forskningsråd | 262695 | Oda Hovet<br>Manuel Carrer<br>Michele Cascella |
| Norwegian Supercomputing Program | NN4654K | Oda Hovet<br>Manuel Carrer<br>Michele Cascella |
| Norges Forskningsråd | 325528 | Nikolina Sekulic |
| National Institute of General Medical Sciences | R35-GM130302 | Jennine Dawicki-McKenna<br>Ben E Black |
| National Institute of General Medical Sciences | GM108360 | Jennine Dawicki-McKenna |

The funders had no role in study design, data collection and interpretation, or the decision to submit the work for publication.

### Author contributions

Dario Segura-Peña, Conceptualization, Data curation, Formal analysis, Supervision, Validation, Investigation, Visualization, Methodology, Writing - original draft, Project administration, Writing - review and editing; Oda Hovet, Formal analysis, Investigation; Hemanga Gogoi, Stine Malene Hansen Wøien, Investigation; Jennine Dawicki-McKenna, Data curation, Investigation, Visualization, Methodology, Writing - review and editing; Manuel Carrer, Visualization; Ben E Black, Conceptualization, Resources, Data curation, Formal analysis, Supervision, Validation, Methodology, Writing - review and editing; Michele Cascella, Resources, Data curation, Formal analysis, Supervision, Validation, Investigation, Visualization, Methodology, Writing - review and editing; Nikolina Sekulic, Conceptualization, Resources, Data curation, Formal analysis, Supervision, Funding acquisition, Validation, Investigation, Visualization, Methodology, Writing - original draft, Project administration, Writing - review and editing

### Author ORCIDs

Dario Segura-Peña (iD) http://orcid.org/0000-0001-6695-1042
Nikolina Sekulic (iD) http://orcid.org/0000-0002-8027-9114

### Decision letter and Author response

Decision letter https://doi.org/10.7554/eLife.85328.sa1
Author response https://doi.org/10.7554/eLife.85328.sa2

## Additional files

### Supplementary files

• Supplementary file 1. Hydrogen-deuterium exchange coupled to mass spectrometry (HDX-MS) summary table for [Aurora B/IN-box]$^{all-P}$ and [Aurora B/IN-box]$^{no-P}$ datasets.

• Supplementary file 2. Statistics of hydrogen-deuterium exchange (HDX) uptake for all peptides over time in [Aurora B/IN-box]$^{all-P}$ and [Aurora B/IN-box]$^{no-P}$ datasets.

• Supplementary file 3. Hydrogen-deuterium exchange coupled to mass spectrometry (HDX-MS) summary table for [Aurora B/IN-box]$^{all-P*}$, [Aurora B/IN-box]$^{no-P*}$, [Aurora B/IN-box]$^{loop-P}$, and [Aurora B/IN-box]$^{IN-P}$ datasets.

• Supplementary file 4. Statistics of hydrogen-deuterium exchange (HDX) uptake for all peptides over time in [Aurora B/IN-box]$^{all-P*}$, [Aurora B/IN-box]$^{no-P*}$, [Aurora B/IN-box]$^{loop-P}$, and [Aurora B/IN-

box]IN-P datasets.

• Supplementary file 5. Full hydrogen-deuterium exchange (HDX) dataset for [Aurora B/IN-box]all-P*
(red), [Aurora B/IN-box]IN-P (purple), [Aurora B/IN-box]loop-P (green), and [Aurora B/IN-box]no-P* (blue).
All collected peptides are shown with the standard error calculated based on two replicates. The
black dot in the corner of the graph indicates the exchange of the fully deuterated control for the
peptide of interest.

• MDAR checklist

## Data availability

All data generated or analysed during this study are included in the manuscript and supporting files.
The mass spectrometry proteomics data are available through the ProteomeXchange Consortium
via the PRIDE partner repository with the dataset identifier PXD038935. Simulation data are freely
available for download at the NIRD repository through the link: https://doi.org/10.11582/2023.00028.

The following datasets were generated:

| Author(s) | Year | Dataset title | Dataset URL | Database and Identifier |
|-----------|------|---------------|-------------|-------------------------|
| Segura-Peña D, Sekulic N | 2023 | The structural basis of the multi-step allosteric activation of Aurora B kinase | https://www.ebi.ac.uk/pride/archive/projects/PXD038935 | PRIDE, PXD038935 |
| Cascella M | 2023 | MD simulations AURKB | https://doi.org/10.11582/2023.00028 | NIRD Research Data Archive, 10.11582/2023.00028 |

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
