## [Editor Report]

This important study investigates the dynamic activation mechanism of a key mitotic kinase complex, Aurora B/INCENP. The method of generating specifically phosphorylated forms of the complex is elegant, supporting a compelling experimental and computational analysis of how these sites synergistically activate Aurora B and providing insight into the dynamics underlying the activation mechanism. This work will be of interest to cell biologists and biochemists studying cell division and kinase regulation.

---

## [Decision Letter]

**Decision letter after peer review:**

Thank you for submitting your article "The structural basis of the multi-step allosteric activation of Aurora B kinase" for consideration by *eLife*. Your article has been reviewed by 3 peer reviewers, one of whom is a member of our Board of Reviewing Editors, and the evaluation has been overseen by Volker Dötsch as the Senior Editor. The following individual involved in review of your submission has agreed to reveal their identity: Richard Bayliss (Reviewer #2).

Essential revisions:

Overall, the reviewers feel quite positively about the manuscript and agree that this work advances our understanding of Aurora B activation. The greatest strength is the strategy to prepare and study enzyme complexes with specific phosphorylation sites occupied, either the activation loop or the TSS sites. The emphasis on changing Aurora B/IN-box dynamics is also a valuable addition to the existing structural models. The primary weakness, detailed by Reviewer 2, is in the MD simulations, and these should be the focus of revisions:

1. It would be instructive to model the interactions of the phosphorylated INCENP with Aurora B using the structure of Aurora C/IN-box (6GR8) as a template. Is it the same or different than the results presented, which start with Aurora B/IN-box (4C2W)?

2. Please compare and contrast the MD models with the Aurora C/IN-box crystal structure, especially regarding the interactions between the IN-box and the activation loop, and hypothesize what could lead to the observed differences.

3. Please discuss the challenges of modeling phosphorylated proteins and how these limitations might affect interpretation of the MD results.

*Reviewer #1 (Recommendations for the authors):*

Reviewer comments from a previous submission were thoroughly addressed, including important controls for unphosphorylated F845C AURKB, continued binding of INCENP to AURKB during HDX, the kinetic contributions of T248 and TSS phosphorylation, and disruption of the AURKB motif by R847 mutation.

To expand more specifically on the point about statistical analysis from the public review:

1. In Figure S1, please indicate the concentration of substrate peptide, the number of replicates, and some measure of experimental variability.

2. In the RMSF plots in Figure 2 and Figure S6, is there a way to statistically test if the distributions are significantly different? The difference is apparent in Figure 2E, but more subtle in Figure 2F, so it would be more convincing with statistical evidence.

3. Please add error bars to Figure S9B.

4. What is the difference between Figure S16B and Figure 5E? Also, please indicate the number of replicates and experimental variability for these figures.

*Reviewer #2 (Recommendations for the authors):*

p.4 EX1 kinetics and EX2 kinetics are technical terms that should be defined and explained.

p.5 Using the 4C2W as a starting point for simulation is justified, as a model for Aurora-B. However, for modelling the interactions of the phosphorylated INCENP, the structure of Aurora-C/pINCENP (6GR8) would be a better option. Because the sequences of Aurora-B and -C are very similar, it would be straightforward to make an Aurora-B/phos-INCENP starting model, which could be "dephosphorylated" in silico to then probe the roles of the different phosphorylation states.

p.6 The comparison of the Aurora-B/IN-box model from MD with the crystal structure of Aurora-C/IN-box is incomplete because there are major differences between them in the IN-box interaction with the kinase activation loop. These should be described and an explanation provided for why the MD simulation leads to a different result from the experimental crystal structure. For example;

– In the Aurora-C/IN-box crystal structure, there are no direct interactions between the IN-box and the phosphate group on the activation loop threonine. In the Aurora-B/IN-box models, either Arg847 or Arg843 forms a direct interaction with the phosphate on the activation loop threonine.

– In the Aurora-C/IN-box crystal structure, the phosphate groups attached to the TSS motif interact directly with Aurora-C, whereas in the Aurora-B/IN-box model they do not, instead they are pointing out into solvent.

– The residues equivalent to Aurora-B/IN-box model Arg847 and Arg843 in the Aurora-C/IN-box structure participate in the interface, but have different interactions. This is important for interpreting the results of mutating these residues because they would be expected to affect binding/activity in either model.

p.9 The rapid formation of very stable salt-bridge interactions between the TSS and adjacent positively charged residues indicates that the modelling parameters may not be optimized for phosphorylated proteins. As these interactions are non-native (i.e. not in the crystal structure of Aurora-C/IN-box), their stability prevents the simulation from reaching the native conformation. Simulation of phosphorylated proteins is challenging, with interactions of the phosphate groups being artificially strong in some standard force field models (e.g. 10.1021/acs.jctc.5b00967, and 10.1021/acs.jcpa.8b04418 for improved polarisable modelling of phosphate). The potential limitations of this aspect of the study should be discussed.

p.10 The loss of activity of the Lys846Asn-Arg847Gln mutant is difficult to understand, what is the rationale? Is this simply a poorly-behaved protein, or how else do these residues contribute to Aurora-B autophosphorylation? Perhaps the individual residue mutations would shed some light on this?

p.11 The sentence mid-way through the first paragraph, that starts "For [Aurora B/IN-box]no-P" is very difficult to understand and is structured with too many commas. It should be rewritten.

p.11 The analysis of global movements is interesting, but the authors should be more cautious about the interpretation because the simulations are of relatively short duration, and the changes may be overly influenced by initial conformational changes as the protein responds to the in silico phosphorylation. The analysis would be improved by including a plot against time/RMSD of the occupancy of the major conformational states.

p.12 Analysis of the EX1-like kinetics suggesting that the helices in the IN-box might be only partially formed whilst the proteins are interacting is very interesting. It would be helpful to the reader to explain this analysis more clearly for a non-expert, perhaps with reference to previous examples in the literature.

p.15 Does one biological replicate really mean a single sample, or does it mean two samples (in which case, two replicates/samples would be more accurate).

p.17 Which software was used for the modelling? Which phosphoryl group model was used (e.g. TP2)?

Supplementary:

p.9 (Figure S5) part C, the phosphoserines are mislabelled pT

p.13 (Figure S8) is there a reason why the quality of one spectrum in each of the spectra sets in Figure S8 is poor?

---

## [Author Response]

Reviewer #1 (Recommendations for the authors):Reviewer comments from a previous submission were thoroughly addressed, including important controls for unphosphorylated F845C AURKB, continued binding of INCENP to AURKB during HDX, the kinetic contributions of T248 and TSS phosphorylation, and disruption of the AURKB motif by R847 mutation.To expand more specifically on the point about statistical analysis from the public review:1. In Figure S1, please indicate the concentration of substrate peptide, the number of replicates, and some measure of experimental variability.

The kinetics shown in Figure S1 are intended to illustrate the previously published differences in the activity of [Aurora B/IN-box]^all-P^ and [Aurora B/IN-box]^no-P^ (Zaytsev, Segura-Peña at al, *eLife*.2016). This measurement was performed several times with high reproducibility. The experimental details of this experiment can now be found in the legend. However, in this work, we carefully determined an initial rate of [Aurora B/IN-box]^all-P^ of 560 ± 10 µM [substrate/min x µM of enzyme] and for [Aurora B/IN-box]^no-P^ of 5 ± 2 µM [substrate/min x µM of Enzyme]. Measurements were performed in triplicate at 600 µM substrate peptide concentration and are shown in Figure 3C.

2. In the RMSF plots in Figure 2 and Figure S6, is there a way to statistically test if the distributions are significantly different? The difference is apparent in Figure 2E, but more subtle in Figure 2F, so it would be more convincing with statistical evidence.

The graph in Figure 2E and 2F plots the probability density for the RMSF values of the specified protein segments. The differences are clear in Figure 2E, and a chi-square test yield a p-value bellow 0.05, indicating that the two distributions are different. The distributions in figure 2F are indeed more similar with a p-value of 0.3. However, it can be noticed in Figure 2F that in the [Aurora B/IN-box]^no-P^ distribution, there is a longer tail towards larger RMSF values, following the same trend with more structural fluctuations and disorder in [Aurora B/IN-box]^no-P^. We have added a sentence to acknowledge that differences are more subtle in the Figure 2F. (Page 7, paragraph 3).

“Although the difference is more subtle in the region of Aurora B that forms the ATP-binding site and is in contact with IN-box^aA^ (Figure 2F) it follows the same trend indicating structural stiffening upon phosphorylation and more disorder in the non-phosphorylated state.”

3. Please add error bars to Figure S9B.

Done.

4. What is the difference between Figure S16B and Figure 5E? Also, please indicate the number of replicates and experimental variability for these figures.

Indeed, in the originally submitted manuscript there was no difference between these two figures. In the present manuscript, we have retained Figure 5E in the main set of figures and added two new panels in the supplement (Figure 5—figure supplement 2B and 2C). In Figure 5E, we show normalized point-by-point data from a 450-min experiment for two partially phosphorylated enzyme forms at three different enzyme concentrations. We now add an additional experiment for the [Aurora B/IN-box]^IN-P^ construct performed in triplicate and with four enzyme concentrations (Figures S16B and C). Figure 5—figure supplement 2B shows the curves with the standard measurement error and Figure 5—figure supplement 2C shows the normalized plots. This experiment reconfirms that phosphorylation of the activation loop Aurora B^Thr248^ occurs in *cis* at the early stages of the activation process. Based on the structural evidence from the crystal structure of Aurora B and Aurora C, the previous report by (Zaytsev, Segura-Peña at al, *eLife*.2016) and the kinetics experiment in this manuscript, we also conclude that phosphorylation in the IN-box occurs in *trans*.

Reviewer #2 (Recommendations for the authors):p.4 EX1 kinetics and EX2 kinetics are technical terms that should be defined and explained.

The reviewer is correct that these terms are technical and therefore difficult to explain. We have now clarified the text that refers to this phenomenon to make it more understandable to a wide audience and to avoid a detailed technical explanation that is beyond the scope of this article (page 4, middle of the last paragraph). In the discussion, we have also expanded the references to examples and reviews that an interested reader can use to find more detailed explanations of this phenomenon. See also the response to comment #8 by this reviewer.

“HDX kinetics for this region shows a strong component of EX1 kinetics and a smaller component of common EX2 kinetics. By definition, EX2 kinetics is common in the native folded state of proteins, while EX1 kinetics are less common in native conditions and is indicative of structural unfolding of local protein regions (Arrington and Robertson, 2000; Chitta et al., 2009; Ferraro et al., 2004; Sperry et al., 2012; Walters et al., 2013). In the IN-box^805-825^ region, we observe the appearance of two clear isotopic envelopes (Figure 1—figure supplement 3), a hallmark of EX1 kinetics. The lower mass isotopic peak corresponds to the folded state of IN-box^805-825^, where the amide-hydrogen bonds are more protected. The high mass isotope peak is indicative of the unfolded state of the IN-box^805-825^ region, where multiple amide-hydrogen bonds break simultaneously and exchange hydrogen for deuterium. The EX1 behavior was observed in both [Aurora B/IN-box]^no-P^ and [Aurora B/IN-box]^all-P^, suggesting that IN-box^805-825^ reversibly switches between the unfolded and folded states regardless of the phosphorylation state. However, the higher mass peak appears more rapidly in [Aurora B/IN-box]^no-P^, suggesting that this region of IN-box visits the unfolded state more frequently in the unphosphorylated form of the enzyme complex. Overall, this observation is consistent with a more disordered IN-box in the unphosphorylated state.”

p.5 Using the 4C2W as a starting point for simulation is justified, as a model for Aurora-B. However, for modelling the interactions of the phosphorylated INCENP, the structure of Aurora-C/pINCENP (6GR8) would be a better option. Because the sequences of Aurora-B and -C are very similar, it would be straightforward to make an Aurora-B/phos-INCENP starting model, which could be "dephosphorylated" in silico to then probe the roles of the different phosphorylation states.

The 4C2W was chosen as the model for our studies because all HDX and kinetic experiments were performed with the same protein, *X. laevis* Aurora B. In addition, one of the monomers in the asymmetric unit was crystallized with the ATP analog, adenosine-5’-[(β,γ)-imido]triphosphate bound in the active site, which we changed only slightly to ATP-Mg before starting the MD simulations. On the other hand, the structure of Aurora C (6GR8) is 75% identical to that of *X. laevis* Aurora B in the structured part and has a bulky inhibitor bound to the active site that is very different from the ATP-Mg. Nevertheless, we created a homology model of *X. laevis* [Aurora B/IN-box] using 6GR8 as template in SwissProt homology modeling software. The model was relaxed in the box of waters and the HADDOCK software was used to dock ATPMg in the active site. As a guide for the final orientation of the ATP-Mg in the active site, we used the highest hits to other Aurora crystals. However, when we ran the MD simulation under the same conditions as before, we found that the ATP-Mg detached from the active site after ~460 ns (Author response image 1). This could be due either to slight inaccuracies in the homology model, a well-documented phenomenon reported previously (see CASP competition reports from past decades), or to incompatibility of the Aurora C structure with the ATP-Mg in the active site. To clarify this ambiguity, we performed MD simulations with the "unmutated" original structure of Aurora C, to which ATP-Mg was similarly docked. Indeed, here, ATPMg was stable in the active site during 1 µs simulation (Author response image 1). Since Aurora B and Aurora C are expected to have the same dynamics, we decided to proceed with [Aurora C/IN-box] for further MD analysis. We observe that during 1 µs simulation, [Aurora C/IN-box]^all-P^ showed no changes in the interactions of the Aurora C^pThr198^ and the phosphorylated TSS motif compared to the original model (crystal structure). The only difference was the flexible confirmation of the six very last C-terminal IN-box residues, which were probably secured in the original conformation by crystal contacts. Next, we removed the phosphates from the TSS motif of IN-box to obtain [Aurora C/IN-box]^loop-P^. Dephosphorylation of INbox leads to a dramatic conformational change of the C-terminal segment of IN-box^886-903^, which leaves its initial position and starts exploring a wide range of conformations. We find the results of this simulation insightful and further supporting our findings of the highly dynamic nature of IN-box. Thus, we now include them in the Discussion, where we discuss the possible reason for the differences between the final conformation observed in our [Aurora B/IN-box]^all-P^ simulation and crystal structure of [Aurora C/INbox]^all-P^ (Page 13, last paragraph).

**Author response image 1. sa2fig1:** Homology model of X. laevis [Aurora B/IN-box] based on the crystal structure of *H. sapiens* [Aurora C/IN-box], [Aurora Bhomology/IN-box]all-P, shows unstable ATP-Mg binding during MD simulation. (**A**) Graph showing root-mean-square deviations (RMSD) for modelled ATP-Mg duringMD simulation in the active site of [Aurora C/IN-box]all-P (magenta) and [Aurora Bhomology/IN-box]all-P (yellow). For Aurora Bhomology/IN-box all-P ATP-Mg RMSD jumps from ~1.5 Å to ~2.75 Å after 440 ns indicating detaching from the active site. In difference, ATP-Mg in [Aurora C/IN-box]all-P remains stabled for full 1000 ns of MD simulation. (**B**) Overlay of the final position of the ATP-Mg in the active site of [Aurora C/IN-box]all-P (magenta) and [Aurora Bhomology/IN-box]all-P (yellow). ATP is shown in sticks and Mg is a ball.

“The results of the MD simulations are consistent with the HDX analysis and highlight the entropic nature of the INbox. However, the C-terminal region of the IN-box adopts a different conformation in our [Aurora B/IN -box]^all-P^ simulation than in the [Aurora C/IN-box]^all-P^ crystal structure, raising the possibility that the observed [Aurora B/IN box]^all-P^ structure is either an intermediate on the activation coordinate or an alternative conformation in the fully phosphorylated state. To better understand the role of phosphorylation on the IN-box conformation, we performed simulations on [Aurora C/IN-box] in the fully phosphorylated state, [Aurora C/IN-box]^all-P^, and in the partially phosphorylated state, [Aurora C/IN-box]^loop-P^, using the [Aurora C/IN-box] crystal structure as a starting model (Figure 7—figure supplement 1). These simulations confirmed that the residues following the TSS motif are highly flexible, even in the fully phosphorylated state. In addition, MD simulations on an [Aurora C/IN-box]^loop-P^ show that the absence of the IN-box phosphates in the [Aurora C/IN-box] structure results in relaxation of the longer C-terminal segment of the IN-box, which contains the TSS-motif. In this simulation, the IN-box is less organized, overall (higher RMSD), than in the [Aurora C/IN-box]^all-P^ simulation and the IN-box^886-903^ rapidly loses association with the phosphorylated activation loop and moves toward the C-lobe of the kinase to adopt a similar, but not identical, conformation to that observed in our original simulation, [Aurora B/IN-box]^loop-P^ (Figure 7—figure supplement 1, inset). The observed behavior confirms that the C-terminal part of the IN-box is a highly entropic unit that follows fast diffusion dynamics directly controlled by the phosphorylation state of the system, with the highest degree of disorder in the unphosphorylated state, partially organized in the [Aurora B(C)/IN -box]^loop-P^ state, and better organized, although still highly dynamic, in the [Aurora B(C)/IN -box]^all-P^.”

In the Results section, we have also clarified that [Aurora B/IN-box]^all-P^ is most likely one of the intermediates along the activation trajectory (see response to the comment #3).

p.6 The comparison of the Aurora-B/IN-box model from MD with the crystal structure of Aurora-C/IN-box is incomplete because there are major differences between them in the IN-box interaction with the kinase activation loop. These should be described and an explanation provided for why the MD simulation leads to a different result from the experimental crystal structure. For example;– In the Aurora-C/IN-box crystal structure, there are no direct interactions between the IN-box and the phosphate group on the activation loop threonine. In the Aurora-B/IN-box models, either Arg847 or Arg843 forms a direct interaction with the phosphate on the activation loop threonine.– In the Aurora-C/IN-box crystal structure, the phosphate groups attached to the TSS motif interact directly with Aurora-C, whereas in the Aurora-B/IN-box model they do not, instead they are pointing out into solvent.– The residues equivalent to Aurora-B/IN-box model Arg847 and Arg843 in the Aurora-C/IN-box structure participate in the interface, but have different interactions. This is important for interpreting the results of mutating these residues because they would be expected to affect binding/activity in either model.

We agree with the reviewer that the contacts in our [Aurora B/IN-box]^all-P^ simulations differ from those in the crystallographic [Aurora C/IN-box]^all-P^ structure, and we have highlighted the differences in the contacts in Figure 2—figure supplement 1 in our original manuscript. As shown in comparable simulations starting with the fully phosphorylated [Aurora C/IN-box] structure, the dynamics of the IN-box are fast and diffusive and responsive to [Aurora B(C)/IN-box] phosphorylation. In our original [Aurora B/IN-box]^allP^ simulation, that starts from partially phosphorylated structure, the IN-box adopts a conformation that likely represents one of the intermediates along the activation coordinate, just as the IN-box conformation in the crystal structure could be one of the several productive conformations. Since our study is intended to illustrate how phosphorylation affects the dynamics of the enzyme complex, rather than focusing on the specific orientation of residues in the active site observed by MD (which cannot yet be performed on time scales long enough to explore the entire conformational space) we refrain from commenting in detail the differences in phosphate interactions, but we clarify in the updated manuscript (page 6, paragraph 4) that the result of [Aurora B/IN-box]^all-P^ simulation most likely represents one of the intermediate steps along the activation coordinate.

“Overall, the structure from our [Aurora B/IN-box]^all-P^ simulation and the [Aurora C/IN-box]^all-P^ crystal structure are very similar. However, an important difference exists in the C-terminal IN-box region (Figure 2—figure supplement 1C). In the [Aurora B/IN box]^all-P^ simulation, the C-terminal region of the IN-box is extended further toward the C-terminal lobe of the kinase and interacts with the phosphorylated activation segment, whereas in the [Aurora C/IN-box]^all-P^ crystal structure there are reciprocal interaction in which arginines from the activation loop interact with the phosphorylated TSS-motif of the IN-box. The latest interactions are not observed in our simulations. We suggest that the crystal structure of [Aurora C/IN-box]^all-P^ is more likely to represent the final active conformational state of the enzyme complex and that the [Aurora B/IN-box]^all-P^ from our simulation is more likely an intermediate state on the autoactivation trajectory. The transition from this intermediate state to the final state observed in the crystal structure could be a relatively slow process that occurs in time spans that we cannot capture with our simulation times. We discuss the possible reason for these differences in more detail in the Discussion.”

p.9 The rapid formation of very stable salt-bridge interactions between the TSS and adjacent positively charged residues indicates that the modelling parameters may not be optimized for phosphorylated proteins. As these interactions are non-native (i.e. not in the crystal structure of Aurora-C/IN-box), their stability prevents the simulation from reaching the native conformation. Simulation of phosphorylated proteins is challenging, with interactions of the phosphate groups being artificially strong in some standard force field models (e.g. 10.1021/acs.jctc.5b00967, and 10.1021/acs.jcpa.8b04418 for improved polarisable modelling of phosphate). The potential limitations of this aspect of the study should be discussed.

We are puzzled by the reviewer's initial comment that the rapid interaction between the phosphorylated TSS motif and the neighbouring arginines is an MD artifact, since the equivalent of the salt bridge INbox^pSer850^-IN-box^Arg843^ is also observed in the crystal structure of Aurora C (6GR8) (Figure 2—figure supplement 1).

Since the crystal structure is only one of the many conformations that the enzyme complex can assume, we also disagree with the argument that observing interactions that are not present in a crystal structure means that the force field we use is not reliable. Moreover, the formation of salt bridges between Aurora B^pThr248^ and arginine from the IN-box instead of salt bridges within the IN-box should be due solely to the topological accessibility of the competing amino acids, since the exchange between a salt bridge with an arginine from the IN-box and one from Aurora B(C) is isoenergetic and thus not affected by the force field parameters themselves. Indeed, in the [Aurora B/IN-box]^loop-P^ simulation, we observe an interaction between Aurora B^Arg843^ and Aurora B^pThr248^, whereas in the [Aurora B/IN-box]^all-P^ simulation, a conformational change of the IN-box led to an isoenergetic interaction between Aurora B^Arg847^ and Aurora B^pThr248^, although both simulations are starting from the same topology.

In general, we agree that force fields, like any other scientific method (computational or experimental), are prone to bias. The reference cited by the reviewer (10.1021/acs.jctc.5b00967) is for interactions with alkali and alkaline earth ions. The development of reliable force fields for main-group metal ions is one of the most difficult and problematic open issues in computer modeling, so the problems cited in this reference are largely due to poor representation of cations. We agree that polarizable models could be a possible improvement for the treatment of electrolytic systems in the future, but their cost-effectiveness compared to traditional force fields is far from proven. At the same time, the CHARMM force field used in this work continues to lead the way in developing the most appropriate parameters for protein simulations, including those for phosphorylated moieties. This force field has been used successfully over several decades in a variety of studies of phosphorylated systems, including ATP-bound proteins and kinases, most notably in a very recent study of Aurora A kinase (DOI: 10.1021/jasms.1c00271). We now add a general commentary on the challenges of MD for phosphorylated proteins in Material and Methods (page 19, paragraph 5).

“Phosphorylation, as one of the most important post-translational modifications, has been thoroughly investigated by computational modelling over the years, in particular with great success for the kinase family (Saladino and Gervasio, 2016). Nonetheless, the involvement of strongly charged, titratable moieties challenge global transferability for the parameters, opening the ground to future improvement based on more sophisticated treatment of electrostatic interactions, like by polarizable force-fields.”

p.10 The loss of activity of the Lys846Asn-Arg847Gln mutant is difficult to understand, what is the rationale? Is this simply a poorly-behaved protein, or how else do these residues contribute to Aurora-B autophosphorylation? Perhaps the individual residue mutations would shed some light on this?

We agree with the reviewer that the loss of activity in this mutant is difficult to understand. Furthermore, we even have kinetic data for a single mutant as part of our original manuscript (see table 1. [Aurora B/INbox^Arg847Ala^]) and this mutant has higher K_M_ for the substrate peptide but similar k_cat_ to the WT enzyme.

Since neither the kinetics results for the [Aurora B/IN-box^Arg847Ala^] single mutant nor the available structures provide a clear explanation to the loss of activity in the [Aurora B/IN-box^Arg846Asn-Arg847gln^] double mutant , we have decide to remove this information from the manuscript.

p.11 The sentence mid-way through the first paragraph, that starts "For [Aurora B/IN-box]no-P" is very difficult to understand and is structured with too many commas. It should be rewritten.

This is now reading (Page 11, middle of the last paragraph):

“However, the type of movement defined by the first principal component (PC1, the one corresponding to the largest relative movement) depends on the phosphorylation status of the enzyme complex. This is open-close movement (for [Aurora B/IN-box]^no-P^), twisting (for [Aurora B/IN-box]^IN-P^), an activation loop movement (for [Aurora B/IN-box]^loop-P^), and synchronized opening and closing of lobes and loop movement (for[Aurora B/IN-box]^all-P^) (Video 5).”

p.11 The analysis of global movements is interesting, but the authors should be more cautious about the interpretation because the simulations are of relatively short duration, and the changes may be overly influenced by initial conformational changes as the protein responds to the in silico phosphorylation. The analysis would be improved by including a plot against time/RMSD of the occupancy of the major conformational states.

We recognize that sampling time is generally an issue for MD based models, but even recent work in the field of Aurora kinases has relied on simulations almost 10 times shorter than those presented here (DOI: 10.1021/jasms.1c00271). However, to confirm that we really observe different structural conformers, we followed the reviewer's suggestion and extended Figure 6—figure supplement 1 to show the change in conformational order parameters as a function of time for the last 1.5 µs of simulations (corresponding to the original Figure 6). The graph clearly shows that the fully phosphorylated system is in each of the three identified conformational basins, and the change between conformations occurs through a sharp transition, confirming that each basin represents a true local minimum of free energy.

Furthermore, the PC analysis of [Aurora C/IN-box]^all-P^ confirmed same dominant PC motions, demonstrating that our PCA is not dominated by initial conformational changes as the protein responds to the *in silico* phosphorylation, but rather are hallmark of global dynamical properties of Aurora kinase that depend only on the topological features of the fold, and not on the specific amino acid sequence (DOI: 10.1021/ja044608+).

p.12 Analysis of the EX1-like kinetics suggesting that the helices in the IN-box might be only partially formed whilst the proteins are interacting is very interesting. It would be helpful to the reader to explain this analysis more clearly for a non-expert, perhaps with reference to previous examples in the literature.

In the Results section (Page 4, middle of the last paragraph; see also first remark from this reviewer) we explained the behavior of EX1-HDX, and here we discuss the implications of this phenomenon in the context of [Aurora B/IN-box] (page 13, paragraph 2). As requested by the reviewer, we have also included references with examples and reviews of EX1 behavior analyzed in the context of other proteins.

“The HDX experiment clearly shows that the IN-box is a highly dynamic polypeptide chain, with most of its regions reaching maximal deuteration after only three minutes of D_2_O incubation (Data set D1), even in the phosphorylated state. Interestingly, the N-terminal part of the IN-box, which is the only part containing defined secondary structure elements, exhibits EX1-like HDX kinetics, suggesting that the IN-box^aA^ and IN-box^aB^ helices reversibly undergo folding and unfolding. Despite the reversible unfolding of the helices, the IN-box remains associated with Aurora B, which was confirmed by our mass photometry experiments (Figure 2 and Figure 1—figure supplement 4). The EX1 kinetics, and its relation to local unfolding of proteins under native conditions, has been described in detail in the literature. (Arrington and Robertson, 2000; Chitta et al., 2009; Ferraro et al., 2004; Krishna et al., 2004; Skinner et al., 2012; Sperry et al., 2012; Walters et al., 2013).»

p.15 Does one biological replicate really mean a single sample, or does it mean two samples (in which case, two replicates/samples would be more accurate).

In HDX experiments, we refer to biological replicates as independent protein preparations and technical replicates as the repeated reads of the same protein preparation. While the biological replicates always follow the same trend, the data cannot be plotted on the same graph due to slight differences in instrumental setup that are very difficult to control. Thus, the error bars we report are due to multiple readings of the same sample (technical replicates). However, we also indicate that the experiment was repeated with different protein preparations and the same result was obtained, which increases confidence in the reported changes.

p.17 Which software was used for the modelling? Which phosphoryl group model was used (e.g. TP2)?

We have now indicated in paragraph 6 on page 18 that MODELLER was used to model 6 missing C-terminal amino acids in the IN -box.

“The ^848^TSSAVWHSP^856^ C-terminal region of IN-box, not visible in the original crystal structure, was modelled using MODELLER (https://www.salilab.org/modeller/) as a random coil.”

Our original manuscript had information on the MD software and the phosphoryl group model.

“The CHARMM36 force-field (Huang et al., 2017) was used to parameterize the protein segments, ATP, and the ions, while water was represented with the triangular TIP3P model (Jorgensen et al., 1983) All bonds involving hydrogen atoms were constrained to their equilibrium distances using the LINCS algorithm (Hess et al., 1997), the LennardJones potentials were computed using a cut-off distance of 10 Å, while the electrostatic interactions were evaluated using the Particle-Mesh Ewald method (Essmann et al., 1995).”

Supplementary:p.9 (Figure S5) part C, the phosphoserines are mislabelled pT

Corrected. Thank you!

p.13 (Figure S8) is there a reason why the quality of one spectrum in each of the spectra sets in Figure S8 is poor?

Figure 3—figure supplement 2 shows the signal for phosphorylated peptides. Panel A shows that the signal for the peptide containing phosphorylated Aurora B^Thr248^ is strong and clearly above the background for [Aurora B/IN-box]^all-P^ and [Aurora B/IN-box]^loop-P^, but for [Aurora B/IN-box]^IN-P^, the signal for this peptide shows low intensity (10^4^ compared to 10^7^ and 10^6^ for the previous peptides) buried in the noise, confirming the absence of phosphorylated Aurora B^Thr248^ in [Aurora B/IN-box]^IN-P^. Panel B shows the peptide with the phosphorylated TSS motif showing a weak signal/absence in [Aurora B/IN -box]^loop-P^.